# Decoupling PER phosphorylation, stability and rhythmic expression from circadian clock function by abolishing PER-CK1 interaction

Yang An[1,2,4], Baoshi Yuan[2,4], Pancheng Xie[2,3], Yue Gu[2], Zhiwei Liu[2], Tao Wang[2], Zhihao Li[2], Ying Xu [2✉] & Yi Liu [3✉]

Robust rhythms of abundances and phosphorylation profiles of PERIOD proteins were thought be the master rhythms that drive mammalian circadian clock functions. PER stability was proposed to be a major determinant of period length. In mammals, CK1 forms stable complexes with PER. Here we identify the PER residues essential for PER-CK1 interaction. In cells and in mice, their mutation abolishes PER phosphorylation and CLOCK hyperphosphorylation, resulting in PER stabilization, arrhythmic PER abundance and impaired negative feedback process, indicating that PER acts as the CK1 scaffold in circadian feedback mechanism. Surprisingly, the mutant mice exhibit robust short period locomotor activity and other physiological rhythms but low amplitude molecular rhythms. PER-CK1 interaction has two opposing roles in regulating CLOCK-BMAL1 activity. These results indicate that the circadian clock can function independently of PER phosphorylation and abundance rhythms due to another PER-CRY-dependent feedback mechanism and that period length can be uncoupled from PER stability.

[1] Model Animal Research Center, Nanjing University, 12 Xuefu Road, Pukou District, Nanjing 210061, China. [2] Cambridge-Su Genomic Resource Center, Soochow University, Suzhou, Jiangsu 215123, China. [3] Department of Physiology, University of Texas Southwestern Medical Center, Dallas, TX 75390, USA. [4]These authors contributed equally: Yang An, Baoshi Yuan. ✉email: yingxu@suda.edu.cn; yi.liu@utsouthwestern.edu

The core eukaryotic circadian oscillators consist of auto-regulatory transcription- and translation-based negative feedback loops[1–5]. In mammals, the heterodimeric CLOCK/BMAL1 complex acts as the positive element by activating the transcription of clock genes by binding to the E-boxes in their promoters[5–7]. PERIOD (PER) and CRYPTOCHROME (CRY) proteins form multimeric complexes that function as the negative elements in the negative feedback loop by inhibiting the CLOCK/BMAL1 activity, which closes the circadian negative feedback loop. Deletion of *mPer1* and *mPer2* in mice completely abolished circadian clock function, indicating they are essential clock components[8,9]. mPER3 does not play a significant role in mice circadian clock function[9]. Functional circadian clocks result in rhythmic expression of clock proteins. Among clock proteins, the abundances of PER1 and PER2 have the most robust circadian rhythms: abundances are very low during the subjective day and high during the subjective night[10]. As with rhythmic expression of negative clock elements in *Neurospora* and *Drosophila*[11,12], rhythmic expression of PER, but not CRY proteins, was proposed to be the master circadian rhythm that is essential for circadian clock function in mammals, and artificial induction of rhythms of PER expression can restore tunable locomotor activity rhythms[13–15].

PER proteins are progressively phosphorylated after their synthesis and become hyperphosphorylated during the late subjective night[10,16]. The combined rhythms of PER abundance and phosphorylation define the time of day during a circadian cycle. Phosphorylation of PER promotes its degradation through the ubiquitin-proteasome pathway mediated by the E3 ligase SCF$^{\beta\text{-TRCP}}$, resulting in decrease of PER levels after late subjective night to allow the reactivation of CLOCK/BMAL1 activity during the subjective day[17,18]. PER phosphorylation has been shown to play critical roles in mammalian circadian clock function by regulation of PER stability, its repressor activity, and its subcellular localization[16,19–21]. The stability of PER appears to correlate with period length: PER in cells with short-period length has increased turnover rate[16,22,23]. Two competing clusters of phosphorylation sites have been shown to act a phospho-switch that affects period length by affecting PER degradation rate[22,24,25]. These results suggest that PER phosphorylation is critical for clock function and that PER stability is a major determinant of circadian period length[23].

From *Neurospora* to mammals, CK1 is the kinase mainly responsible for phosphorylation of FRQ and PER proteins[21,24,26–29]. Unlike typical kinase-substrate interactions, which are weak and transient, CK1 forms tight stoichiometric complexes with FRQ and with PER in *Neurospora*, *Drosophila*, and mammals[10,30–36]. This conserved feature suggests a common mechanism in eukaryotic clocks. The PER phosphorylation events by CK1 were previously examined by in vitro assays in the absence of stable PER-CK1 association[20,24,37,38]. In vivo, the stable PER-CK1 association can potentially have a dramatic effect on PER phosphorylation efficiency due to high CK1 local concentration. By creating FRQ point mutations that specifically abolished stable FRQ-CK1 interaction, we previously showed that the FRQ-CK1 interaction is essential for clock function because it is required for both the CK1-mediated FRQ phosphorylation and the negative feedback mechanism[26,39–41]. These results demonstrate that the negative feedback process is mediated by FRQ-dependent CK1 phosphorylation of WC proteins, which represses their activity[26,41–44]. In addition, the strength of the FRQ-CK1 interaction but not FRQ stability is the main determinant of circadian period length in *Neurospora*[41,45]. In mammals, the PER-dependent CLOCK phosphorylation was first suggested by using clock protein complexes purified from liver samples[32]. More recently, CK1δ and the casein kinase binding domains are

shown to be required for the removal of CLOCK-BMAL1 complex from promoter E-boxes in cells[46]. In addition, a CK1 binding domain in *Drosophila* was previously shown to be critical for PER phosphorylation by CK1, transcription repressor activity and circadian clock function[47,48].

CRY proteins are essential for circadian rhythmicity in adult animals and are important for robust cellular rhythms[49–51]. In addition, the repressor function of PER has been shown to require CRY proteins in cellular studies[46,52]. However, circadian rhythms can be observed in neonatal SCN of *Cry* double deficient mice and the loss of circadian rhythms in adult mice was proposed to be due to desynchronization of cellular rhythms[53–55], suggesting that circadian negative feedback mechanism can function independently of CRY proteins under certain conditions. Although genetic evidence indicates that PER proteins function as main negative elements in the mammalian circadian negative feedback loop, how they function to repress CLOCK-BMAL1 activity is not well understood. Our understanding of the circadian negative feedback mechanism is also complicated by sometimes conflicting results between in vitro and in vivo studies. Despite the essential role of PER in the circadian negative feedback loop based on genetic studies, CRY proteins were found to bind to the CLOCK–BMAL1–E-box complex directly. Moreover, the expression of CRY alone in cells is sufficient to repress E-box-mediated transcription driven by CLOCK-BMAL1 probably by sequestering BMAL1 from transcription coactivators CBP/P300, whereas expression of PER in cells has little or no effect[56–60]. In addition, the inhibition of CLOCK–BMAL1 by PER is CRY dependent in cells and was proposed to cause the removal of the CRY-CLOCK-BMAL1 complex from E-boxes in mice[52,61]. Based on these results, two types of negative feedback mechanisms were proposed: the CRY-dependent repression of CLOCK-BMAL1 activity on DNA and the PER and CRY-dependent removal of CLOCK-BMAL1 complex from E-boxes[52]. More recently, PER was proposed to repress CLOCK-BMAL1 DNA binding activity by promoting PER-dependent CK1 phosphorylation of CLOCK[46]. However, since most of the studies concerning the circadian negative feedback mechanism have been done in cells in culture, the mechanism in vivo remains unclear.

In this study, we demonstrate that PER act as the CK1 scaffold to promote hyperphosphorylation of CLOCK protein, which closes the circadian negative feedback loop by removing CLOCK-BMAL1 complex from DNA. In addition, the PER-CK1 interaction has two opposing roles in regulating CLOCK-BMAL1 activity. Importantly, our results also indicate that the circadian clock can still function independently of PER phosphorylation and PER abundance rhythms due to another PER-CRY-dependent feedback mechanism and that circadian period length can be uncoupled from PER stability. Our conclusions represent a revision to the current circadian clock model and provide critical insights to the mammalian clock mechanism.

## Results

**Identification of hPER2 residues required for stable hPER2-CK1δ association.** As expected, immunoprecipitation assays showed that human PER1-3 can stably associate with CK1δ when they were individually transiently expressed with CK1δ in HEK293T cells but the interaction between hPER2 and CK1 appeared to be the strongest (Fig. 1a). A Casein kinase 1 binding domain of mouse PER2 (amino acids 555-754) was previously identified to be important for PER phosphorylation by CK1 in cells[35]. In addition, mPER2 internal deletion assays identified two 25aa motifs within this domain that are critical for CK1-mediated PER2 phosphorylation and its turnover by β-TrCP[18]. However, the exact residues essential for PER association with CK1 are not

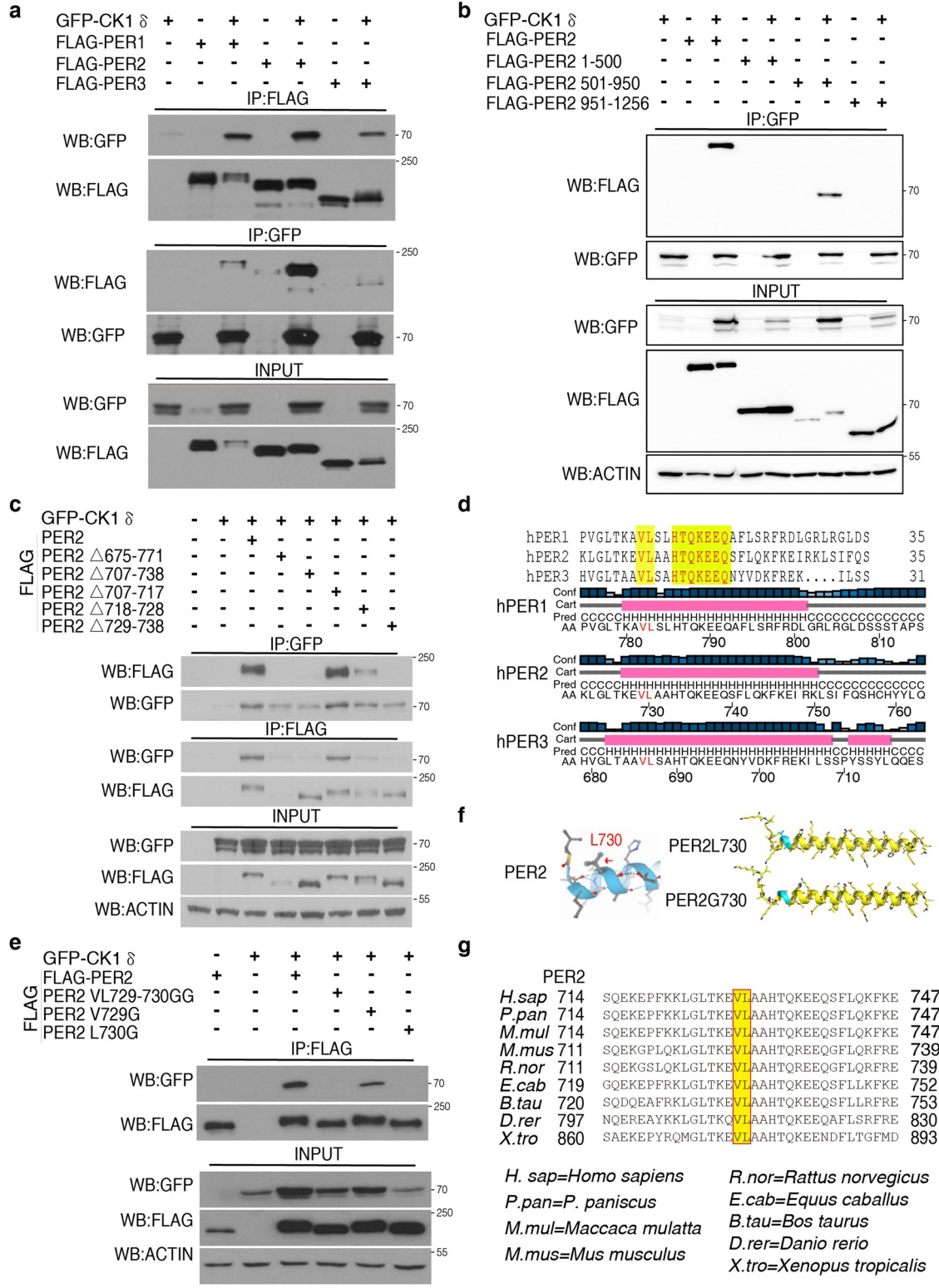

known. To identify the hPER residues required for hPER-CK1δ association, we transiently expressed different regions of hPER2 with CK1δ in HEK293T cells and showed that amino acids 501-950 of hPER2, but not other regions, can associate with CK1δ (Fig. 1b). A series of hPER2 internal deletions showed that the amino acids 729-738 of hPER2 are required for hPER2-CK1δ association (Fig. 1c and Supplementary Fig. 1a–c). This region of hPER2 is conserved in hPER1, hPER3, other vertebrate PER homologs and overlaps with one of the motifs previously identified

to be critical for mPER2 turnover by β-TrCP (Fig. 1d, e)[18]. In addition, this domain is also conserved in *Drosophila* and its deletion was previously shown to be critical for PER phosphorylation by CK1, transcription repressor activity and circadian clock function[47,48]. Such conservations indicate a common circadian clock mechanism in animals.

Protein structure prediction using alphafold (https://alphafold.ebi.ac.uk/) revealed that this region is predicted to form an α helix in all three hPER proteins (Fig. 1f). The requirement for α helix

**Fig. 1 Identification of the PER2 domains and residues required for PER2-CK1δ interaction. a–c** Western blot analyses of immunoprecipitation assays of FLAG-tagged PER constructs and GFP-tagged CK1δ were expressed in HEK293 cells (3 μg of PER plasmid and 2 μg of CK1δ plasmid transfected per 60-mm dish of cells) and showed that CK1δ associates with hPER when co-expressed in HEK293 cells. The levels of PER2 proteins and CK1δ were determined by western blot using anti-FLAG and GFP antibody, respectively. Analyses of the interaction of FLAG-tagged PER1, PER2, and PER3 with GFP-tagged CK1δ (**a**). Identification of the region of hPER2 sufficient for the hPER2-CK1δ interaction (**b**). Analyses of hPER2 deletion constructs (**c**). Three independent experiments were performed to validate the results. See also Fig. S1a–c. **d** Amino acid sequence alignment of PCD domains of hPER1-3 proteins. Yellow regions mark conserved residues in the α helical domain (top). Diagrams showing the secondary structure prediction of the hPER proteins (bottom). **e** Western blot analyses of immunoprecipitation assays of FLAG-tagged PER constructs with the indicated mutation and GFP-tagged CK1δ were expressed in HEK293 cells. Three independent experiments were performed to validate the results. **f** Structure prediction of the wild-type hPER2 (top right) and hPER2 (V729G-L730G) (bottom right) region described in 1d by Alphafold. The L730 residue in the hPER2 was indicated by a red arrow in an enlarged picture on the left. **g** Amino acid sequence alignment of the PCD domains of nine indicated vertebrate PER2 homologs.

domain for clock protein association with CK1 and the presence of leucine residues in this domain are reminiscent of the requirement for an α helix in FRQ for its association with CK1 in *Neurospora*[26]. To examine this possibility, we mutated hPER2 valine 729 and leucine 730 individually or simultaneously to glycine residues. These two residues are located at the end of a long α helix. Structure prediction by alphafold indicated that the glycine mutations do not affect the α helix formation of this region (Fig. 1f). Immunoprecipitation assays showed that the V729G mutation was reduced and that the single L730G and V729G-L730G double mutations completely abolished the hPER2-CK1 association (Fig. 1e and Supplementary Fig. 1d). These two residues are also conserved in this domain of different vertebrate PER2 homologs (Fig. 1g). These data indicate that the α helix in hPER2 that includes V729 and L730 is required for stable interaction with CK1. We refer to this α helix of PER as the PER-CK1 docking (PCD) site. It should be noted that we consistently observed reduced CK1 levels in the input samples when CK1 was expressed alone or co-expressed with hPER2 mutants that cannot interact with CK1 (Fig. 1b, e), suggesting that the PER-CK1 interaction can stabilize CK1.

**Stable PER2-CK1δ interaction is required for PER2 phosphorylation and its degradation by CK1δ.** We examined the role of stable association of PER-CK1δ on PER phosphorylation by CK1δ in cells. Co-transfecting hPER2 and CK1δ into HEK293T cells resulted in robust hPER2 phosphorylation as indicated by a mobility shift (Fig. 2a). In contrast, the deletion of amino acids 729-738, the single L730G mutation, and the V729G-L730G double mutation abolished the CK1δ-induced hPER2 mobility shift (Fig. 2a, b). In addition, treatment with lambda phosphatase suggested that the loss of the stable hPER2-CK1δ interaction abolished detectable hPER2 phosphorylation by CK1δ in cells. Because of the roles of PER phosphorylation in regulating its stability, we compared the degradation rates of hPER2 and hPER2(L730G) after the addition of protein synthesis inhibitor cycloheximide. The hPER2(L730G) protein was much more stable (half-life ~ 4 h) than was wild-type hPER2 (half-life ~1 h) (Fig. 2c).

To confirm that the stable hPER2-CK1δ association is required for the CK1-mediated PER2 phosphorylation at sites known to be physiologically important, we examined the CK1δ phosphorylation of hPER2 on serine 662. A mutation at this site leads to familial advanced sleep phase syndrome[16,19,20,24]. Western blot analysis using an antibody that specifically recognizes PER2 phosphorylated at S662 showed that CK1δ induced phosphorylation of S662 of wild-type hPER2 but not of PER with S662G/S662D mutations in HEK293T cells, indicating the specificity of the antibody (Fig. 2d). As predicted, CK1δ failed to phosphorylate S662 of hPER2 with the L730G mutation. Together, these results indicate that in order for CK1δ to phosphorylate hPER2 in cells,

it must first dock stably at the PCD site. These results also indicate that although CK1δ is responsible for phosphorylation of many sites on PER proteins, these sites by themselves are not optimal CK1 phosphorylation sites and their phosphorylation in cells requires a high local concentration of CK1, which is achieved by the stably PER-CK1 association at the PER PCD site.

The stable association of CK1 with PER proteins also raises the possibility that PER functions as a scaffold protein that mediates CK1 phosphorylation of other proteins in the PER complex. To test this hypothesis, we determined whether CK1 can induce phosphorylation of hPER2(L730G) phosphorylation in presence of hPER1. Indeed, the co-expression of PER1 with hPER2(L730G) resulted in the phosphorylation of hPER2(L730G) (Fig. 2e). The mutation of the corresponding PCD site in mPER1 abolished this effect (Supplementary Fig. 2). In addition, although CK1δ did not bind directly to hPER2(L730G), the co-expression of PER1 led to co-immunoprecipitation of CK1δ with hPER2(L730G) (Fig. 2e). This result indicates that PER1 recruits CK1 to phosphorylate PER2(L730G) in the PER1-PER2 complex (Fig. 2f). Thus, PER proteins act as a scaffold for CK1 to promote phosphorylation of proteins in the PER-containing protein complex.

**PER2 acts as a scaffold to induce hyperphosphorylation of CLOCK by CK1δ.** The stable PER and CK1 association and ability of PER to act as CK1 scaffold prompted us to examine whether PER can promote phosphorylation of CLOCK to close the circadian negative feedback loop in a manner similar to FRQ-mediated CK1 phosphorylation of WC proteins in *Neurospora*. Previous studies have shown that the various forms of CLOCK observed in mobility shift assays are due to phosphorylation events[62,63]. The expression of wild-type hPER2 in HEK293T cells that also express CLOCK resulted in hyperphosphorylation of CLOCK, but the expression of hPER2(L730G) did not (Fig. 3a), indicating that PER2 can indeed act as a CK1 scaffold to promote CLOCK phosphorylation (Fig. 3b). In addition, the CK1-dependent hyperphosphorylation of CLOCK is dependent on the concentration of hPER2 as CLOCK phosphorylation increased as the hPER2 expression level increased (Fig. 3c).

Purification of the mouse CLOCK-containing protein complex and mass spectrometry analyses previously identified serines at residues 38, 42, 427, and 431 as CLOCK phosphorylation sites, and the phospho-mimic mutations at these sites resulted in the reduction of CLOCK-BMAL1 E-box activation activity[62,63]. To examine whether these CLOCK phosphorylation events are dependent on the PER-scaffolding function, we created two CLOCK protein mutants, M2 (S427G/S431G/S436G/S437G/S440G/S441G) and M3 (S38G/S42G/S427G/S431G/S436G/S437G/S440G/S441G), in which these phosphorylation sites and putative sites immediately downstream were mutated to glycine residues. The co-expression of PER and CK1δ failed to induce

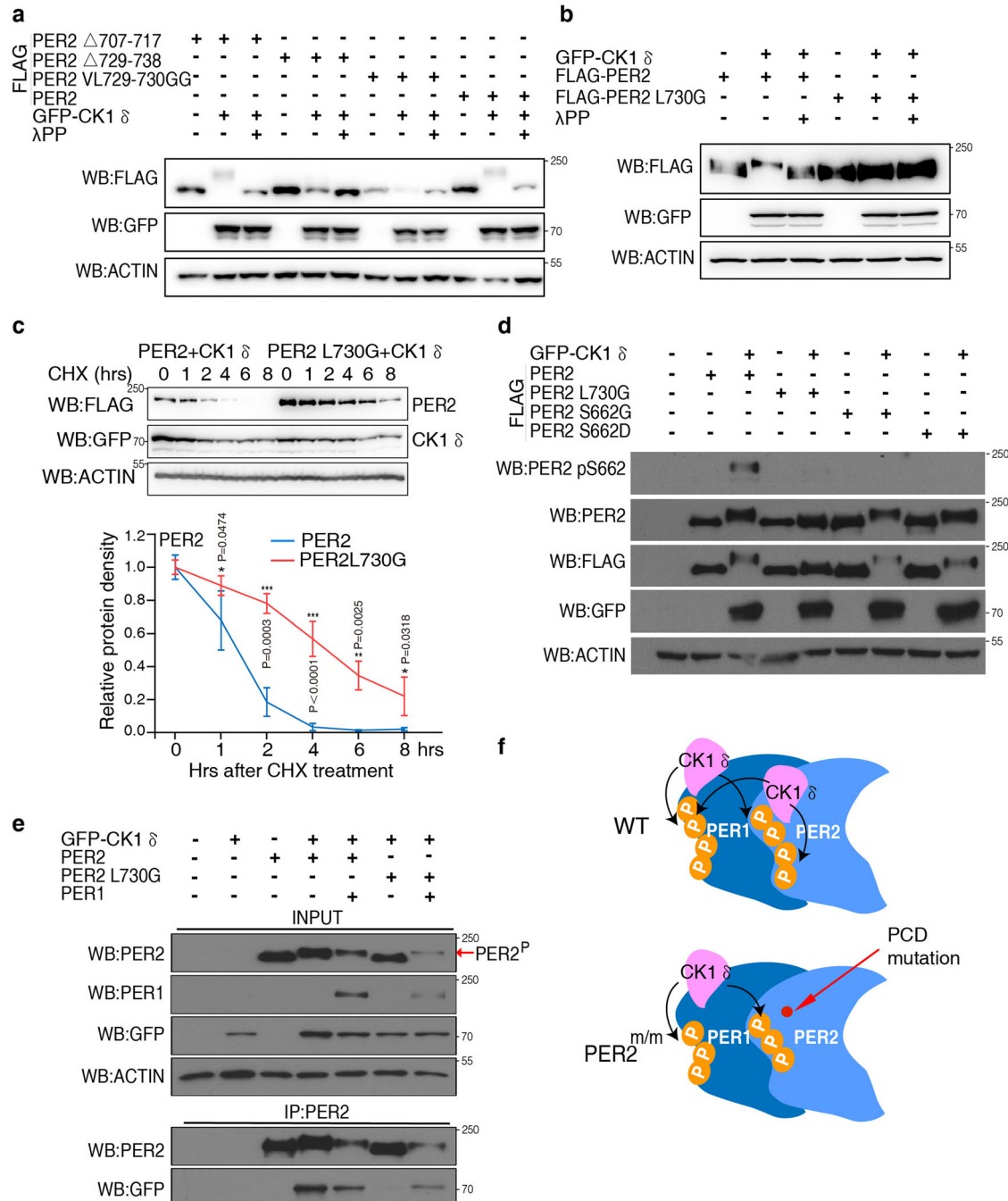

**Fig. 2 The stable CK1δ-PER interaction is required for hPER phosphorylation by CK1δ and promotes PER degradation. a, b** V729G/L730G or L730G mutations of hPER2 abolishes PER2 phosphorylation by CK1δ in cells. Upshift of hPER2 mobilities indicate its phosphorylation by CK1δ. Indicated samples were treated with lambda phosphatase to confirm the mobility shifts were due to phosphorylation. 1.5 μg of hPER2 and 1 μg of CK1δ plasmids were transfected per 35 mm dish of HEK293 cells. The low PER2 levels in lane 5 and 7 are due to variations in transfect experiments. Three independent experiments were performed to validate the results. **c** CHX treatment showing that the L730G mutation dramatically slows down hPER2 turnover rate. Western blot result of a representative experiment showing the levels of hPER2 after the addition of CHX (10 μg/ml) (top). ($n = 3$). Densitometric analysis of three independent experiments of the Western blot results (bottom). * and *** indicate $p$ value < 0.05 and 0.001 (two-sided $t$-test), respectively, at the indicated time points. Data are represented as mean ± SD. **d** Western blot analysis showing the hPER2 L730G mutation abolished hPER2 S662 phosphorylation. Three independent experiments were performed to validate the results. **e** Immunoprecipitation assays and western blot analysis showing that hPER2[L730G] can be phosphorylated by CK1δ when hPER1 was co-expressed. The co-expression of hPER1 results in immunoprecipitation of hPER2[L730G] with CK1δ. The phosphorylated hPER2 species are indicated by the arrow. See also Fig. S2. Three independent experiments were performed to validate the results. **f** A diagram showing the model that CK1 uses PER as a scaffold to phosphorylate different PER proteins in the complex.

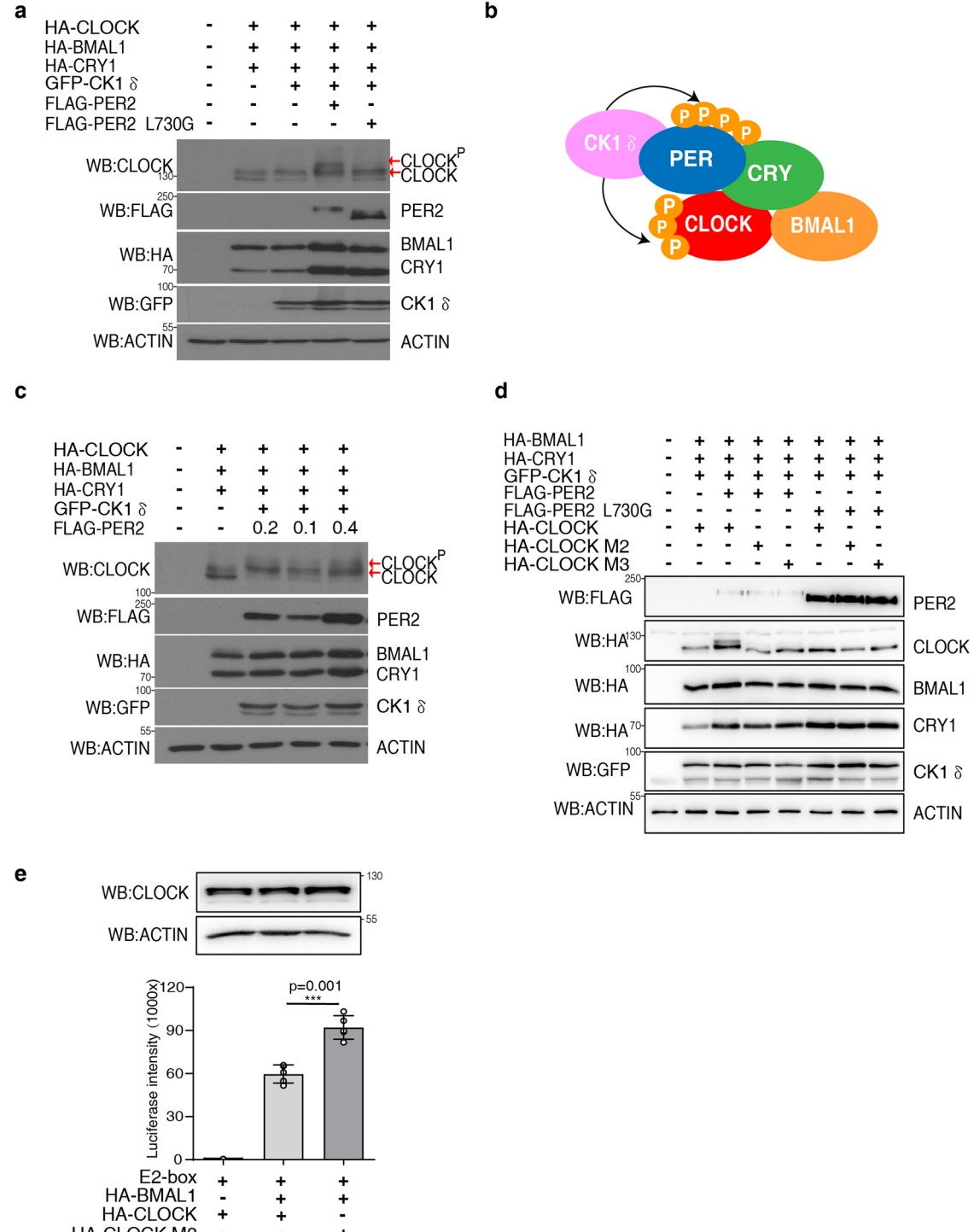

**Fig. 3 The PER-CK1 interaction is required for CLOCK hyperphosphorylation which can repress CLOCK-BMAL1 activity. a** Western blot analysis showing that hPER2 expression results in hyperphosphorylation of CLOCK but L730G mutation of hPER2 blocks CLOCK hyperphosphorylation. CLOCK, BMAL1, CRY1 and CK1δ expression plasmids were cotransfected with PER2 or PER2L730G in HEK293 cells. 200 ng of BMAL1, 20 ng of CRY1,400 ng of CLOCK, 400 ng of CK1δ and 400 ng of PER2 or PER2L730G plasmids per 35 mm dish of HEK293 cells. The hyperphosphorylated CLOCK was indicated by the top arrow. Note that the endogenous CLOCK protein was not detected due to its low expression level. Five independent experiments were performed to validate the results. **b** A diagram depicting the model that PER2 acts as the CK1 scaffold to promote the phosphorylation of PER and CLOCK proteins in the PER-CRY-CLOCK-BAML1 complex. **c** Hyperphosphorylation of CLOCK is PER expression dose-dependent. The indicated different amounts of hPER2 expression plasmids were used in the transfection. The hyperphosphorylated CLOCK was indicated by the top arrow. Three independent experiments were performed to validate the results. **d** Mutation of CLOCK phosphorylation sites abolished the PER-dependent CLOCK hyperphosphorylation. The previously identified phosphorylation CLOCK sites were mutated in the M1 (S427/431/436/437/440/441 to glycine) and M2 (S38/42/S427/431/436/437/440/441 to glycine) mutants. Three independent experiments were performed to validate the results. **e** Mutations of CLOCK phosphorylation sites result in increased expression of the CLOCK/BMAL1-driven E-box controlled luciferase reporter gene. ***p value < 0.001 (two-side t-test). Data are represented as mean ± SD, n = 5. The western blot results of the same experiment were shown above.

hyperphosphorylation of M2 and M3 (Fig. 3d, lane 4–5), suggesting that these sites are required for CLOCK hyperphosphorylated mediated by PER-recruited CK1. Furthermore, consistent with previous results[62,63], promoter E-box activity assays showed that expression of the M3 CLOCK mutant resulted in an increased CLOCK-BMAL1 transcription activation activity (Fig. 3e). Consistent with our conclusion, Cao et al. reported that CK1δ and the casein kinase binding domains are required for the removal of CLOCK-BMAL1 complex from promoter E-boxes in cells[46]. In addition, the PER- and CK1-dependent CLOCK hyperphosphorylation results in the reduction of CLOCK-BMAL1 E-box binding activity in vitro. Together, our results suggest a conserved circadian negative feedback mechanism from fungi to mammals in which FRQ and PER, respectively, act as scaffolds for CK1 to promote WC and CLOCK phosphorylation, which represses the activities of the circadian positive elements.

**Mice with PER2 lacking the ability to stably interact with CK1 have constitutive unphosphorylated mPER2 and hypophosphorylated CLOCK.** To determine the physiological function of PER2-CK1 stable association in vivo, we generated a knock-in mouse strain that expresses mPER2 with the V720G/L721G double mutation, corresponding to the human hPER2 with V729G/L730G mutation that abolished the hPER2-CK1 association. The *mPer2* knock-in mouse *Per2*$^{V72G0/L721G}$ (referred to as *Per2*$^{m/m}$) was generated using a CRISPR-Cas9 approach (Fig. 4a, b). The *PER2*$^{m/m}$ mice were morphologically normal and fertile with the expected Mendelian ratio. Because the complete loss of circadian rhythms requires deletion of both *mPer1* and *mPer2* genes[8,9], we generated the *Per1*$^{-/-}$; *Per2*$^{m/m}$ double mutant mice. To determine the in vivo impact of the mutations on mPER2-CK1δ complex formation and mPER2 phosphorylation, we prepared protein lysates from mouse liver tissues collected at circadian time 4 h and 20 h (CT4 and CT20, respectively) from wild-type (WT), *Per1*$^{-/-}$; *Per2*$^{m/m}$, and *Per2*$^{m/m}$ mice and performed immunoprecipitation using an antiserum specific for the endogenous mPER2. As predicted, the association between CK1δ and mPER2$^{V720G/L721G}$ was completely abolished in the *Per1*$^{-/-}$; *Per2*$^{m/m}$ mice (Fig. 4c), demonstrating that the PCD site is essential for the stable PER-CK1 complex formation in vivo. In addition, mPER2$^{V720G/L721G}$ was hypophosphorylated at both time points in the *Per1*$^{-/-}$; *Per2*$^{m/m}$ mice. In contrast, the association with CK1 was maintained and mPER2$^{V720G/L721G}$ was phosphorylated in the *Per2*$^{m/m}$ single mutant mice (Fig. 4c), indicating that although the mutation in mPER2 abolished its interaction with CK1δ, mPER1 recruited CK1 to the PER1-PER2 complex to mediate mPER2 phosphorylation by CK1.

To determine the impact of the loss of mPER2-CK1 interaction on mPER2 phosphorylation, we treated the protein lysates obtained at CT20 from WT and *Per1*$^{-/-}$; *Per2*$^{m/m}$ liver tissue with lambda protein phosphatase. Phosphatase treatment of the WT sample converted phosphorylated mPER2 forms into one form that migrated faster in the gel (Fig. 4d). Side-by-side comparison of the mPER2 proteins from the WT and *Per1*$^{-/-}$; *Per2*$^{m/m}$ samples showed that the dephosphorylated mPER2 from the WT sample and the untreated mPER2$^{V720G/L721G}$ had identical gel mobility (Fig. 4d). Phosphatase treatment failed to change gel mobility of the PER2$^{V720G/L721G}$ protein. Furthermore, the mPER2 gel mobility in the *Per1*$^{-/-}$; PER2$^{m/m}$ double and *Per1*$^{-/-}$; *Per2*$^{m/m}$; *Per3*$^{-/-}$ triple mutants were identical (Fig. 4e), confirming that mPER3 plays a negligible role in promoting PER phosphorylation by CK1. These results confirm our results in HEK293 cells and demonstrate that mPER2-CK1 complex formation is required for detectable mPER2 phosphorylation events in vivo in the *Per1*$^{-/-}$; PER2$^{m/m}$ mice. Thus,

in vivo, a stable PER-CK1 complex must be formed before PER can be phosphorylated by CK1 and possibly other kinases.

We then compared expression profiles of clock proteins in the WT and *Per1*$^{-/-}$; *Per2*$^{m/m}$ liver tissues at different time points over a circadian cycle. As expected, there were robust rhythms of both amounts and phosphorylation profiles of mPER2 in the WT mice: PER levels were low and the protein was hypophosphorylated during the subjective day, and levels were high (peaking at CT20) and the proteins became progressively phosphorylated during the subjective night (Fig. 4f, g). In addition, CLOCK phosphorylation was rhythmic and became hyperphosphorylated at CT16-20, which are time points corresponding to the peak mPER2 levels and low CLOCK-BMAL1 binding at E-boxes[64]. BMAL1 also had a phosphorylation profile rhythm but with an opposite phase from that of the mPER2 rhythm: BMAL1 was hyperphosphorylated during the day and became hypophosphorylated when mPER2 peaked. In the *Per1*$^{-/-}$; *Per2*$^{m/m}$ liver, however, all these rhythms were abolished (Fig. 4f, g): mPER2 levels were constantly high and the protein was not phosphorylated, consistent with the dramatically increased PER stability. Confirming the role of mPER2 in promoting CLOCK phosphorylation by CK1 in vivo, in the *Per1*$^{-/-}$; *Per2*$^{m/m}$ livers, CLOCK was constantly hypophosphorylated, and BMAL1 was constantly hyperphosphorylated. The role of PER in promoting CLOCK phosphorylation but inhibiting BMAL1 phosphorylation is consistent with results from a previous study in which both *mPer1* and *mPer2* genes were deleted[46]. Thus, our data confirm that PER acts a CK1 scaffold to promote CLOCK hyperphosphorylation in vivo.

To confirm the loss of both mPER2 phosphorylation and the abundance rhythm of mPER2 levels in the *Per1*$^{-/-}$; *Per2*$^{m/m}$ mice, we examined nuclear mPER2 expression profiles. The absence of α-TUBULIN in the nuclear samples indicates that the nuclear samples were free of cytoplasmic protein contamination (Supplementary Fig. 3). The levels of nuclear PER2$^{V720G/L721G}$ were constant and the protein was not phosphorylated at different circadian time points (Fig. 4h, i), confirming that mPER2 phosphorylation and its rhythmic expression requires the PER-CK1 interaction. In addition, this result also indicates that the CK1-dependent PER phosphorylation is not required for PER nuclear entry in vivo.

**The loss of PER-CK1 association affects period length but does not abolish circadian locomotor activity rhythms.** To determine the impact of the loss of mPER2 phosphorylation, mPER2 abundance rhythm, and the PER-dependent CLOCK hyperphosphorylation on circadian rhythms, we examined locomotor rhythms of the mutant mice. The *mPer2*$^{m/m}$ single mutant mice exhibited a locomotor rhythm period that was more than 3 h longer (27.1 ± 0.38 h) than that of their WT littermates (23.7 ± 0.1 h) (Fig. 5a, b), confirming that the mPER2-CK1 interaction is important for clock function and period determination. Unexpectedly, the *Per1*$^{-/-}$; *Per2*$^{m/m}$ mice exhibited robust wheel-running activity rhythms in light-dark cycles (LD) and in constant darkness (DD): The rhythms could be synchronized by LD cycles and exhibited robust free-running activity rhythms in DD. In DD, unlike the long period rhythms of the *mPer2*$^{m/m}$ single mutant, the locomotor rhythm period of the *Per1*$^{-/-}$; *Per2*$^{m/m}$ mice was ~1.4 h shorter than that of the WT mice (Fig. 5a, b). Although mPER3 does not play a significant role in the mouse circadian clock[9,35,65], we obtained the *mPer3*$^{-/-}$ mice (Supplementary Fig. 4) and the *Per1*$^{-/-}$; *Per2*$^{m/m}$; *Per3*$^{-/-}$ triple mutant mice to exclude the possibility that mPER3 can compensate for the loss of mPER1. As expected, the triple mutant

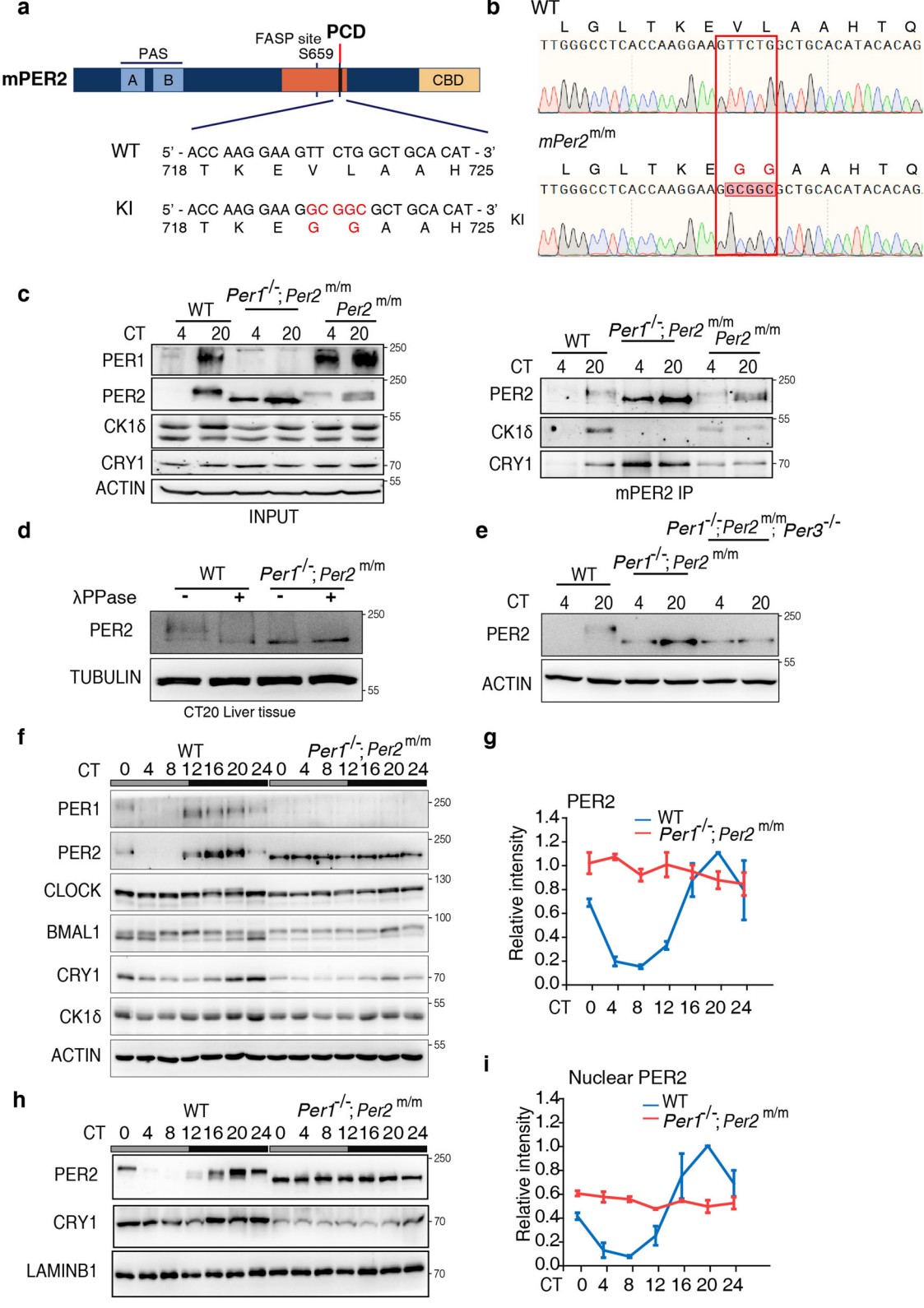

mice also had robust short-period locomotor activity rhythms as the $Per1^{-/-}$; $Per2^{m/m}$ double mutant (Fig. 5a, b).

To further determine the impact of the loss of mPER2-CK1 interaction on circadian rhythms of other physiological processes, we monitored circadian rhythms of metabolism and activity of mice in metabolic cages. As shown in Fig. 5c, the rhythms of

oxygen consumption rate (VO2), respiratory exchange ratio (RER) and activity were comparable in the wild-type and $Per1^{-/-}$; $Per2^{m/m}$ mice. Thus, robust circadian rhythms of physiological processes can still be maintained in mutant mice.

Together, these results indicate that although the PER2-CK1 association is important for circadian period determination, it is

**Fig. 4 Constitutive unphosphorylated PER2 and hypophosphorylated CLOCK in the mPER2 PCD knock-in mice. a** A diagram depicting domains of mPER2 protein and the knock-in (KI) mutations made in the mPER2 PCD domain. **b** Comparison of the DNA sequencing results in the wild-type (WT) and the $Per2^{m/m}$ mice. **c** Immunoprecipitation assays and Western blot results showing the levels and phosphorylation profiles of mPER2 and other proteins in the liver extracts of the indicated mouse strains. mPER2 antibody was used in the immunoprecipitation assays. Three independent experiments were performed to validate the results. **d** Western blot results showing the comparison of mPER2 phosphorylation profiles of the liver extracts collected at CT20 of the WT and $Per1^{-/-}$; $Per2^{m/m}$ mice. The protein samples were treated with lambda phosphatase. Three independent experiments were performed to validate the results. **e** Western blot results comparing the mPER2 phosphorylation profiles of the liver extracts collected at CT20 of the WT, $Per1^{-/-}$; $Per2^{m/m}$ and $Per1^{-/-}$; $Per2^{m/m}$; $Per3^{-/-}$ mice. Three independent experiments were performed to validate the results. **f** Western blot results showing the levels and phosphorylation profiles of mouse clock proteins at the indicated time points in constant darkness. Three independent experiments were performed to validate the results. **g** The quantification of mPER2 levels of Western blot results in the WT and $Per1^{-/-}$; $Per2^{m/m}$ mice. Data are represented as mean ± SD, $n = 3$. **h** Western blot results showing the levels and phosphorylation profiles of the nuclear mPER2 protein at the indicated time points in constant darkness. See also Supplementary Fig. 3. **i** The quantification of the nuclear mPER2 levels of Western blot results in the WT and $Per1^{-/-}$; $Per2^{m/m}$ mice. Data are represented as mean ± SD, $n = 3$.

---

not required for circadian clock function. The circadian clock can still function when PER phosphorylation, PER abundance rhythm, and the negative feedback mechanism mediated by the PER-dependent CLOCK hyperphosphorylation by CK1 are lost. The specific elimination of PER-CK1 interaction in the mutant mice, thus, led to the decoupling of circadian clock functions from the circadian oscillations of PER levels and its phosphorylation. Furthermore, the short-period locomotor phenotypes of both $Per1^{-/-}$; $Per^{m/m}$ and $Per1^{-/-}$; $Per2^{m/m}$; $Per3^{-/-}$ mutant mice despite dramatically increased PER2 stability indicate that circadian period length can also be uncoupled from PER stability.

**The PER2-CK1δ interaction is required for robust circadian gene expression in mouse liver: most CCGs become arrhythmic but some remain rhythmic with low amplitudes.** To examine circadian gene expression genome-wide at molecular level, we performed RNA-seq analysis of liver samples obtained every 4 h over a 24-h period for the WT and $Per1^{-/-}$; $Per2^{m/m}$ mice in DD. Using the MetaCycle package (with parameter cycMethod = c("JTK")), we identified 4203 rhythmically expressed transcripts in the WT liver; this was 23% of all quantified transcripts. In the $Per1^{-/-}$; $Per2^{m/m}$ mice, however, most of these clock-controlled genes (CCGs) became arrhythmic but 670 (15.9%) remained rhythmic (Fig. 6a, b). A comparison of the Metacycle analyses using a period of 24 and 22.3 h found that 95% of identified rhythmic genes in the 24 h period analysis are also found to be rhythmic when a period of 22.3 h was used. These results indicate that despite the robust locomotor activity rhythms of the mutant mice, the disruption of the PER-CK1 interaction abolished the rhythmic expression of the vast majority of clock-controlled genes (CCGs). Importantly, for the shared CCGs in both mice strains, the amplitudes of most of their transcript rhythms were severely dampened in the $Per1^{-/-}$; $Per2^{m/m}$ mice (Fig. 6b). To quantitatively validate this conclusion, we calculated the amplitudes of the circadian rhythms of mRNAs of the shared CCGs in both strains[64,66]. Their comparison confirmed that the rhythm amplitudes were severely reduced in the mutant mice (Fig. 6c). Consistent with these results, the transcripts of many of the core clock genes such as $Per2$, $Bmal1$, $Clock$, $Cry1$, $Dbp$, and $Nr1d1$ (also known as $Rev$-$erbα$) were still rhythmic but with severely reduced amplitudes in the $Per1^{-/-}$; $Per2^{m/m}$ mice (Fig. 6d). The low amplitude $Per2$ mRNA rhythm but arrhythmic mPER2 protein is likely due to the dramatically increased mPER2 protein stability in the $Per1^{-/-}$; $Per2^{m/m}$ mice. Our RNA-seq experiment to examine rhythmic gene expression was limited to one circadian cycle, which may affect our characterization of the number of rhythmically expressed genes.

These results demonstrate that although the PER-CK1 interaction is required for the circadian rhythms of most clock-controlled

genes, circadian gene expression does occur in the absence of this interaction, albeit with severely reduced amplitudes. The severely damped circadian gene expression in the liver but the robust locomotor activity rhythms of the $Per1^{-/-}$; $Per2^{m/m}$ and $Per1^{-/-}$; $Per2^{m/m}$; $Per3^{-/-}$ mutant mice suggest that the latter is due to neuronal network effects of the suprachiasmatic nucleus[51,53–55,67], which allows neurons to synchronize to permit robust locomotor activity rhythms despite severely impaired clock function.

**The PER2-CK1 interaction is required for efficient removal of CLOCK-BMAL1 from E-boxes.** To determine the role of PER2-CK1 interaction in promoting the removal of CLOCK-BMAL1 complex from DNA in vivo, we performed BMAL1 chromatin immunoprecipitation (ChIP) assays of mouse liver tissues collected at different time points over a circadian cycle. As expected, the BMAL1 binding at the E-boxes of $Dbp$, $Nr1d1$, $Cry1$, and $Per1$ exhibited robust circadian rhythms with peaks at CT8-12 and troughs around CT16-20 in the WT mice (Fig. 6e, f). In the $Per1^{-/-}$; $Per2^{m/m}$ mice, BMAL1 E-box rhythms were also observed, but amplitudes were reduced, consistent with the low amplitude molecular rhythms (Fig. 6e, f). Although the BMAL1 E-box enrichment levels during the subjective day were similar in both WT and mutant mice, levels were markedly higher during the subjective night in the mutant mice than in the WT animals (Fig. 6e, f). Further, in the mutant mice, troughs were observed at CT16 that were phase advanced from those of the WT mice (Fig. 6e, f), consistent with the short-period phenotype of the mutant. These results confirm the role of the PER-CK1 interaction in promoting the removal of CLOCK-BMAL1 from E-boxes. However, this function appears to be subjective night specific, which corresponds to the time of high PER levels and CLOCK hyperphosphorylation in the wild-type mice. The BMAL1 E-box binding rhythms at the promoters of these clock genes further confirm that a circadian clock with a near 24-h period can function independently of PER phosphorylation, PER protein oscillation, and the negative feedback mechanism mediated by PER-dependent CLOCK phosphorylation by CK1. Thus, there is a CK1-independent negative feedback mechanism that is sufficient to generate circadian rhythms with a period close to 24 h, albeit with low amplitudes.

**PER-CK1 interaction decreases inhibition of CLOCK-BMAL activity on DNA.** The transcription of $Dbp$ and $Nr1d1$ is mostly due to transcriptional activation by CLOCK-BMAL1 through their binding to E-boxes in the promoters of these genes, and PER expression results in their repression in a CRY-dependent manner[52,61,64,66,68]. Thus, levels of $Dbp$ and $Nr1d1$ mRNAs were used as reporters of CLOCK-BMAL1 activity at E-boxes. With the increased association of the CLOCK-BMAL1 complex

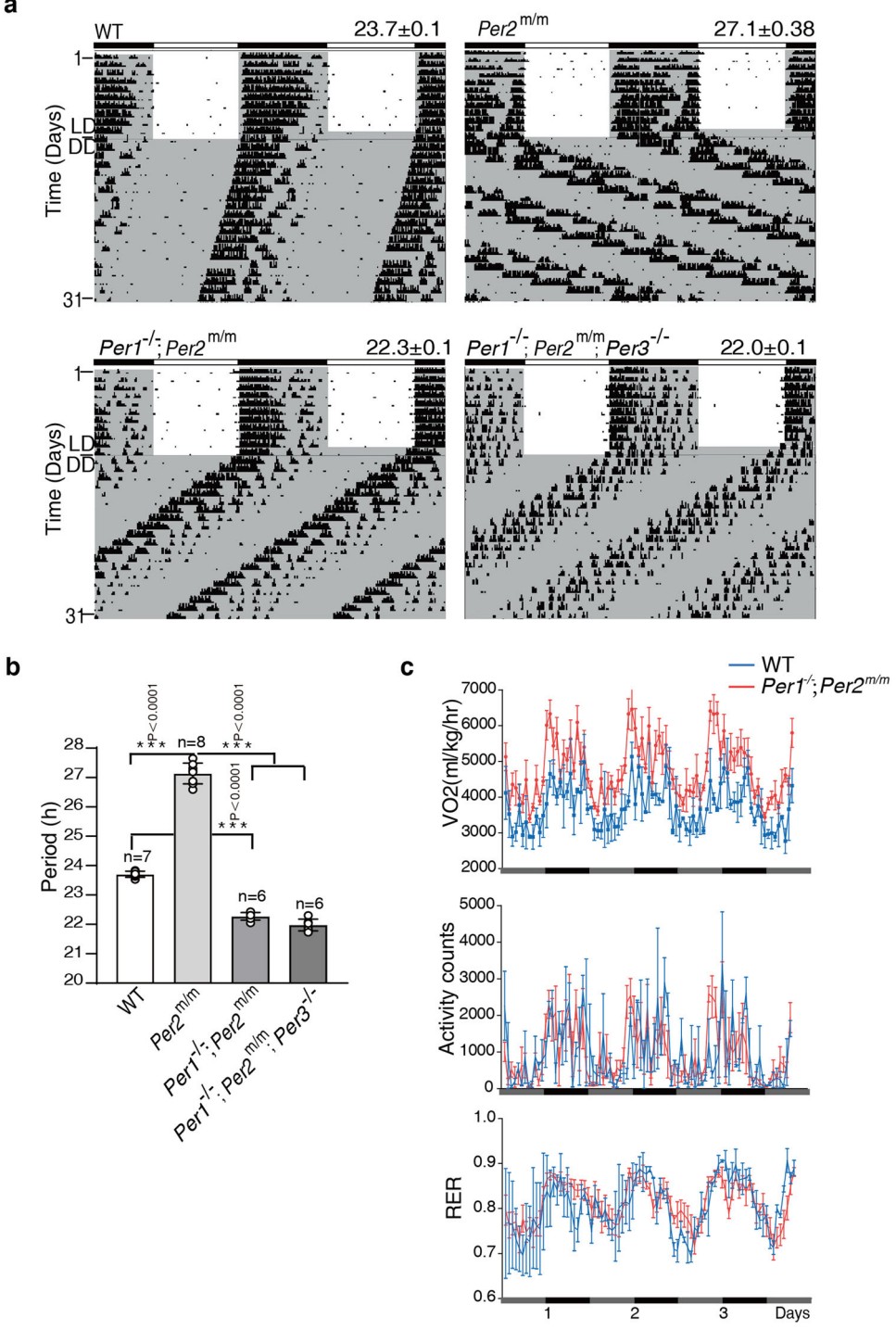

**Fig. 5 Loss of PER-CK1 interaction alters circadian period length but does not abolish robust circadian behavior and other physiological rhythms.**
**a** Locomotor activity recordings of representative mice for the WT, *Per1*−/−; *Per2*m/m and *Per1*−/−; *Per2*m/m; *Per3*−/− mice. Alternative white and dark bars indicate the LD cycles used for entrainment prior to release into constant darkness. See also Supplementary Fig. 4. **b** Average period length (±SD) quantification of the indicated mouse strains by Clocklab. The indicated *p* values were determined using Student's *t*-test (two-sided). Error bars are standard deviations. **c** Circadian rhythms of oxygen consumption rate, respiratory exchange ratio and activity in the wild-type and *Per1*−/−; *Per2*m/m mice in a comprehensive lab animal monitoring system in constant darkness. *n* = 5 per genotype. Error bars are SEMs.

at the E-boxes in the *Per1*−/−; *Per2*m/m mice, we expected to see an increase of *Dbp* and *Nr1d1* mRNA levels. Surprisingly, however, the low amplitude rhythms of *Dbp* and *Nr1d1* transcripts oscillated near their trough levels, and their peak levels were much lower in the mutant mice than in the WT mice (Fig. 6d), indicating that the CLOCK-BMAL1 activity was constantly

inhibited on DNA in the mutant mice despite the high CLOCK-BMAL1 levels at the E-boxes. These results suggest that PER has two different roles in the negative feedback process: promoting the removal of CLOCK-BMAL1 complex from DNA via PER-dependent CLOCK hyperphosphorylation by CK1 and inhibition of CLOCK-BMAL1 activity on DNA by unphosphorylated PER.

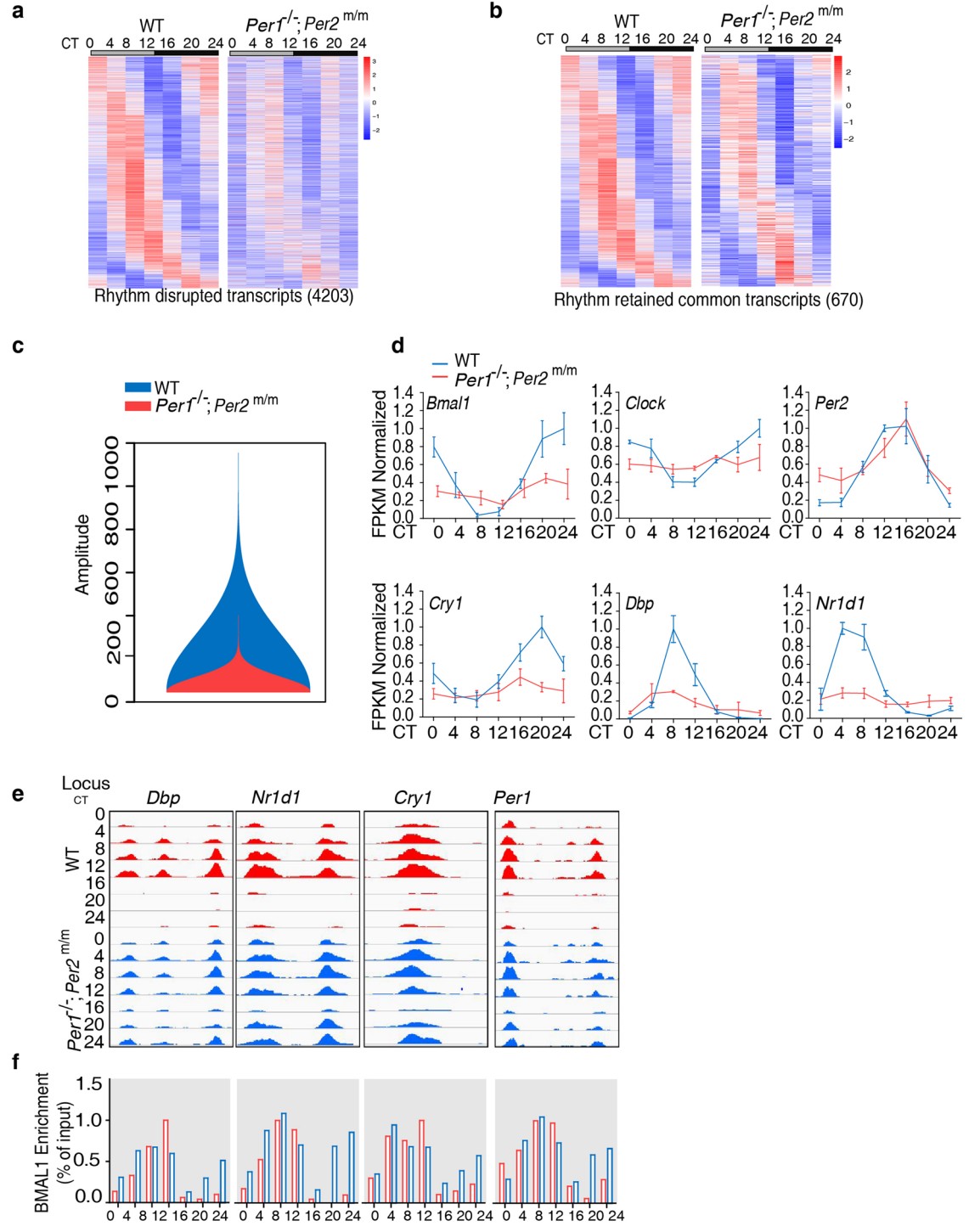

**Fig. 6 Disruption of CK1δ-PER interaction impairs robust rhythmic gene expression and removal of CLOCK-BMAL1 complex from E-boxes during subjective night. a** Heatmap of transcript levels of the identified 4203 rhythmic genes in the WT and the corresponding genes in the $Per1^{-/-}$; $Per2^{m/m}$ mice at the indicated circadian time points. RNA-seq was performed using liver tissue collected at the indicated time points ($n = 3$ per time point). The color bar indicates the scale used to show the expression levels of transcripts normalized to Z-scores. Rhythmic expressed genes were identified by meta2d function in MetaCycle with adjusted $p < 0.05$, period 24 h. **b** Heatmap of common 670 rhythmic genes in the WT and $Per1^{-/-}$; $Per2^{m/m}$ mice (MetaCycle package, $P < 0.05$). The color bar indicates the scale used to show the expression of transcripts across seven time points, with expression normalized to Z-scores. **c** Violin plot comparing the amplitudes of common rhythmic genes in the WT and $Per1^{-/-}$; $Per2^{m/m}$ mice. The relative amplitudes of the 670 common rhythmic genes in the WT and $Per1^{-/-}$; $Per2^{m/m}$ mice were obtained from MetaCycle package and the violin plot was generated using R package vioplot (https://cran.r-project.org/web/packages/vioplot/vioplot.pdf). **d** Rhythmic transcript levels of selected clock genes in the WT and $Per1^{-/-}$; $Per2^{m/m}$ mice determined by RNA-seq. Data are represented as mean ± SD, $n = 3$. **e** BMAL1 ChIP-Seq results at the indicated time points and loci in the WT and $Per1^{-/-}$; $Per2^{m/m}$ mice. UCSC genome browser views of BMAL1 occupancy are shown. Each track represents the normalized ChIP-seq read coverage (wiggle plot) across regulatory regions of the indicated genes. **f** Quantification of relative BMAL1 levels at the indicated E-box loci at the different time points.

The strong inhibition of CLOCK-BMAL1 activity in the $Per1^{-/-}$; $Per2^{m/m}$ mice provides an explanation for its short-period phenotype.

The circadian rhythms in the $Per1^{-/-}$; $Per2^{m/m}$ mice suggest that the unphosphorylated mPER2 can mediate a functional circadian negative feedback mechanism together with CRY proteins. Expression of CRY has been shown to repress CLOCK-BMAL1 activity at E-boxes without removing the complexes from DNA[56,61,69]. In addition, PER has been proposed to stabilize CRY by preventing its ubiquitination and degradation[10,13,70,71]. However, despite the high PER2 level, the nuclear CRY1 protein levels are constantly low in the $Per1^{-/-}$; $Per2^{m/m}$ mice (Fig. 4f, h), which should be mainly due to the low Cry1 mRNA levels (Fig. 6d). These results suggest that the strong inhibition of the CLOCK-BMAL1 activity in the $Per1^{-/-}$; $Per2^{m/m}$ mice is not due to high CRY1 levels.

Immunoprecipitation of mPER2 proteins using mouse liver tissue samples showed that the amount of CRY1 associated with mPER2 was constantly high in the $Per1^{-/-}$; $Per2^{m/m}$ mice (Fig. 7a), suggesting unphosphorylated mPER2 has a strong ability to form a complex with CRY1. We performed CRY1 ChIP assays of liver tissues of WT and $Per1^{-/-}$; $Per2^{m/m}$ mice at different time points in DD to examine the association of CRY1 at E-boxes. As expected, CRY1 enrichment at different E-boxes exhibited a circadian rhythm in the WT mice with a peak at CT0 and a trough around CT16 (Fig. 7b). In the $Per1^{-/-}$; $Per2^{m/m}$ mice, such rhythms were abolished and CRY1 enrichment at E-boxes were constant at the intermediate WT level even though the nuclear CRY1 level was constantly low (Figs. 4f and 7b). These results suggest that the inhibition of the CLOCK-BMAL1 transcription activation activity in the $Per1^{-/-}$; $Per2^{m/m}$ mice is likely due to the strong PER-CRY association that can promote CRY1 association at the E-boxes and enhances its repressor activity. In addition, it is also likely that CK1 docked on PER proteins may also phosphorylate PER-associated CRY to reduce its repressor activity[71–73]. Thus, in addition to promote the CK1-dependent CLOCK-BMAL1 removal from DNA, PER also contribute to the circadian negative feedback process by regulating the inhibitory activity of CRY independent of CK1.

The two opposing effects of PER-CK1 interaction on CLOCK-BMAL1 activity (promoting CLOCK-BMAL1 removal from E-boxes but decreasing CRY-mediated inhibition of CLOCK-BMAL1 activity on DNA) provide an explanation for the modest impact on E-box reporter activity when PER is expressed in cells[56–60]. Together, these results indicate that despite the loss of the negative feedback mechanism mediated by the PER-dependent CK1 phosphorylation of CLOCK, unphosphorylated PER and CRY can still sustain residual clock functions and allows a functional circadian negative feedback loop to function. Therefore, optimal mammalian clock functions require the coupling of both the CK1-dependent and CK1-independent negative feedback loops (Fig. 7c).

## Discussion
The formation of stable stoichiometric complexes between CK1 isoforms and PER/FRQ proteins is a conserved feature in eukaryotic circadian clock mechanisms. By mutating the amino acid residues required for the PER-CK1 interaction, our study allowed us to specifically understand the role of PER-CK1 interaction in PER phosphorylation and the PER-dependent functions of CK1 in the circadian negative feedback mechanism. Our experiments in cells and in the mice demonstrated that the stable complex formation between PER2 and CK1 is required for detectable PER phosphorylation in vivo. These results are in

contrast with in vitro kinase assays using recombinant CK1 and short PER peptides, in which CK1 does not form a tight complex with its substrate[20,24,37,38]. Our results demonstrate that CK1 must first tightly dock on PER before it can phosphorylate various PER phosphorylation sites in vivo and in cells. The formation of the PER-CK1 complex results in a high local concentration of CK1 which allow non-optimal PER sites to be phosphorylated. In addition to CK1, CK2 and glycogen synthase kinase-3 have also been shown to be PER kinases[74,75]. The complete loss of phosphorylated PER species in the $Per1^{-/-}$; $Per2^{m/m}$ mice suggests that phosphorylation by CK1 is also required for PER phosphorylation by other kinases.

We established that PER acts a scaffold for CK1 to promote phosphorylation of different PER proteins and of CLOCK. In the $Per2^{m/m}$ single mutant mice, mPER2 was phosphorylated due to its association with mPER1. Both in cells and in mice, PER2 promotes hyperphosphorylation of CLOCK. Furthermore, the association of CLOCK-BMAL1 at E-boxes of clock-controlled genes were markedly increased in the $Per1^{-/-}$, $Per2^{m/m}$ mutant mice during subjective night. Thus, our results in cells and in vivo demonstrated that the CK1 hyperphosphorylation of CLOCK mediated by the PER-CK1 interaction is a major negative feedback mechanism in the mammalian clock. This mechanism results in the efficient removal of CLOCK-BMAL1 complexes from E-boxes. This conclusion is consistent with recent in vitro and cellular results that suggest that the PER-dependent CK1 phosphorylation of CLOCK results in the removal of CLOCK-BMAL1 complex from E-boxes[46] and indicates that this circadian negative feedback mechanism is conserved from fungi to mammals[26,42,44,76]. The functional importance of this negative feedback mechanism in the mammalian clock function was demonstrated by the loss of rhythmic gene expression for the vast majority of CCGs in the $Per1^{-/-}$, $Per2^{m/m}$ mice. The 27 h long period of the $Per2^{m/m}$ single mutant mice is likely caused by reduced kinetics of CLOCK phosphorylation and/or removal of the CLOCK-BMAL1 complex from DNA due to the inability of mPER2$^{m/m}$ to bind CK1 even though PER2/CLOCK phosphorylation can still be mediated by PER1-associated CK1.

Unlike CLOCK, BMAL1 was constantly hyperphosphorylated in the $Per1^{-/-}$; $Per2^{m/m}$ and $Per1^{-/-}$; $Per2^{-/-}$ mice[46], indicating that PER and PER-CK1 interaction inhibit BMAL1 phosphorylation. It was previously shown that BMAL1 is phosphorylated by Casein kinase (CK)−2α and CRY mediates periodic binding of CK2β to BMAL1 to inhibit BMAL1 phosphorylation by CK2α[77,78]. It is likely that PER and PER-CK1 interactions are also required for the inhibitory role of CRY in BMAL1 phosphorylation.

In the $Per1^{-/-}$, $Per2^{m/m}$ mutant mice, in addition to the loss of PER phosphorylation, the circadian rhythm of PER abundance was also abolished, resulting in constant high PER levels due to dramatically increased PER stability. Although rhythmic expression of PER was proposed to be essential for circadian clock function in mammals[13–15], the $Per1^{-/-}$; $Per2^{m/m}$ mutant mice exhibit robust circadian locomotor activity and other physiological rhythms despite the loss of PER phosphorylation and arrhythmic protein levels. Moreover, low amplitudes of circadian rhythms of gene expression and CLOCK-BMAL1 association on E-boxes were observed for many clock-controlled genes including some core clock genes. Thus, the circadian clock still functions, albeit at much reduced amplitude when PER phosphorylation, PER abundance rhythm, and the PER-CK1-mediated negative feedback mechanism were abolished. Constitutive overexpression of wild-type PER2 in the brain of transgenic mice was previously shown to result in the loss of locomotor activity rhythms[13]. Unlike the mPER2$^{V720G/L721G}$ protein, which cannot interact

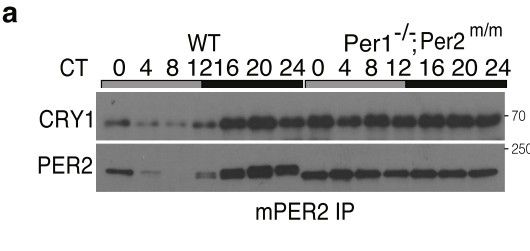

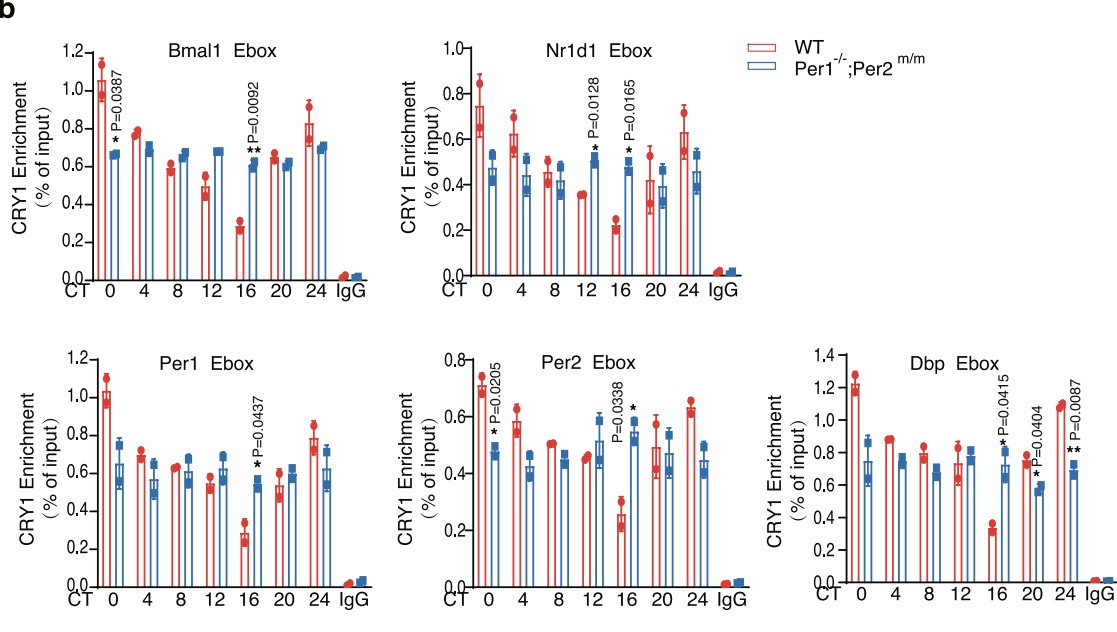

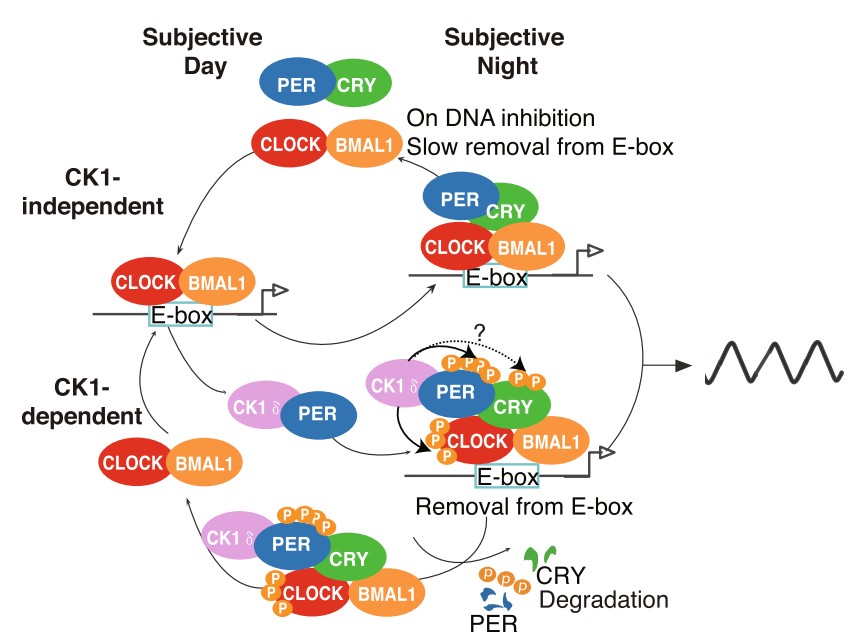

**Fig. 7 Disruption of the CK1δ-PER interaction increases CRY1 enrichment at the E-box region. a** Immunoprecipitation assay using mPER2 antibody showing the levels of CRY1 associated with mPER2 in the WT and $Per1^{-/-}$; $Per2^{m/m}$ mouse livers. Three independent experiments were performed to validate the results. **b** CRY1 ChIP assay results showing the relative CRY1 enrichment at the E-box region of the indicated gene promoters at the indicated time points in the WT and $Per1^{-/-}$; $Per2^{m/m}$ mice. Chromatin samples from mouse livers were analyzed by ChIP using CRY1 antibody. Error bars are standard deviations. *$p$ value < 0.05 and **$p$ value < 0.01 (two-sided $t$-test). **c** A model of the coupled CK1-dependent and CK1-independent negative feedback loops in the core mammalian circadian oscillator. During subjective days, the PER levels are low and CLOCK-BMAL is associated with E-box and active. During the subjective night, PER and CRY levels are high. The CK1-dependent negative feedback process (bottom) involving CK1, PER, and CRY that mediates efficient removal CLOCK-BMAL1 from the E-boxes by promoting PER-dependent phosphorylation of CLOCK by CK1. The CK1-independent process (top) results in the repression of the CLOCK-BMAL1 complex transcription activation activity by the PER-CRY complex on DNA. This process also causes slow removal of CLOCK-BMAL1 complex from E-boxes.

with CK1, the exogenous WT mPER2 protein can associate with CK1, which can result in constant removal of CLOCK-BMAL1 from the E-boxes.

Because PER proteins are essential for function of the mammalian clock, our results indicate that the mammalian clock oscillator has two coupled circadian negative feedback loops: One is dependent on a complex involving CK1, PER, and CRY that mediates efficient CLOCK-BMAL1 removal from the E-boxes during subjective night by promoting PER and CRY-dependent CLOCK phosphorylation. The other is CK1-independent which causes repression of the transcription activation activity of the CLOCK-BMAL1 complex by the PER-CRY complex on DNA (Fig. 7c). The low amplitude but rhythmic CLOCK-BMAL1 association with E-boxes in the $Per1^{-/-}$; $Per2^{m/m}$ mice suggests that the latter mechanism can also result in the slow removal of the CLOCK-BMAL1 complex from E-boxes.

The mRNA levels of the CLOCK-BMAL1-mediated expression of E-box driven genes were markedly reduced despite the increased BMAL1 association on E-boxes in the $Per1^{-/-}$, $Per2^{m/m}$ mice. This result indicates that the CK1-dependent mechanism decreases CLOCK-BMAL1 transcription activation activity. In addition, we also found that although CRY1 protein levels are low in the $Per1^{-/-}$; $Per2^{m/m}$ mice, the amount of CRY associated with PER2 is constantly high, suggesting that unphosphorylated PER has a strong ability to form a complex with CRY. The strong inhibition of CLOCK-BMAL1 activity in the $Per1^{-/-}$; $Per2^{m/m}$ mice is likely due to the strong PER-CRY association which may enhance the CRY repressor activity. In addition, CK1 associated with PER may phosphorylate PER-associated CRY, which can regulate its repressor activity[71–73].

The results presented here indicate that PER and CRY can sustain a functional negative feedback loop by repressing CLOCK-BMAL1 activity independently of the PER-CK1 interaction. Due to the loss of PER phosphorylation and PER abundance rhythm in the $Per1^{-/-}$; $Per2^{m/m}$ mice, another uncharacterized rhythmic event should be responsible for the rhythmic suppression of CLOCK-BMAL1 activity by PER-CRY to achieve near wild-type period length. The importance of this CRY-dependent circadian negative feedback loop is highlighted by the importance of CRY stability and activity in period length determination[71,79–81].

On the other hand, clear circadian rhythms of gene expression and neuronal firing could be observed in neonatal SCN tissues of the $Cry$ double deficient mice and the loss of circadian rhythms in adult mice was proposed to be due to desynchronization of cellular rhythms[53–55]. Thus, the PER-CK1-dependent negative feedback mechanism may be also capable of supporting circadian gene expression at the cellular level independent of CRY proteins. The long period rhythms of the $Per2^{m/m}$ mice indicate that this feedback process has an important role in period determination. In the absence of the PER-CK1 association, the PER and CRY-dependent mechanism results in an increased inhibition of CLOCK-BMAL1 activity, which may result in the short-period phenotype in the $Per1^{-/-}$; $Per2^{m/m}$ mice. Therefore, although each of two modes of CLOCK-BMAL1 inhibition by PER and CRY can sustain a functional circadian negative feedback loop, these mechanisms are coupled together to achieve robust and physiologically relevant circadian clock functions.

## Methods

**Mouse strains and behavioral analysis**. Animal studies were conducted in the accredited SPF animal facility in the CAM-SU Genomic Resource Center and approved by the Animal Care and Use Committee of Soochow University, Suzhou, China (APYX2018-2, APYX2018-6). PER2$^{V721G/L722G}$ mice were generated by CRISPR/Cas9 technology on C57BL/6 J background through the import of a mutant homologous repair template in the region corresponding to amino acids 721-722 site. The targeted guide RNA, Cas9 mRNA, and repair template were mixed and injected into two-cell stage embryos of C57BL/6 J mice, and the embryos were then transplanted into a surrogate mouse uterus. The founders were characterized by genotyping and then crossed to C57BL/6J mice. $Per3$ knockout was made by the CAM-SU Genomic Resource Center. $Per1$ knockout mice was obtained from Jackson laboratory[8]. The F1 offspring were analyzed by DNA sequencing to confirm the germline transmission of intended mutations and then crossed to C57BL/6J, $Per1^{-/-}$; or $Per3^{-/-}$ mice. The relevant sequences and primers are provided in source data file at Fig. 4b.

For locomotor phenotype screening, mice older than 8 weeks were transferred to cages with running wheels and housed alone to examine their locomotor activity profiles. After synchronization in 12 h–12 h light-dark cycle for at least 7 days, mice were transferred into DD and free-running activity the free-running period by Clocklab as previously described[82].

**Metabolic cage experiments**. Mice were entrained to LD12/12 cycles for 1 week. After 3 days of the acclimation period, the mice were transferred to constant dark in the comprehensive lab animal monitoring system (Oxymax; Columbus Instruments)[83]. The VO2, RER and activity of mice were continuously recorded every 20 min for 4 days in DD. The results were normalized and were plotted as the mean ± SEM ($n = 5$ per genotype).

**Cell culture, plasmid transfection, degradation experiment, and western blots**. HEK293T cells were grown in Dulbecco's modified eagle medium (DMEM)/high glucose with 10% fetal bovine serum (FBS) as well as 50 U/ml penicillin and 50 μg/ml streptomycin. All cell lines were maintained under standard conditions (37 °C, 5% CO$_2$). Deletion or point mutant hPer2 sequences were cloned by PCR and inserted into pCMV-Tag 2B (Promega) and confirmed by DNA sequencing. Lipofectamine 3000 (Invitrogen) and polyethylenimine were used for plasmid transfection according to the vendor protocol.

To determine hPER2 stability, PER2 and CK1δ plasmids were transfected into HEK293T cells. After 36 h, the cells were treated with 10 μg/ml CHX (Sigma). Transfected cells were collected from 0 to 8 h after CHX treatment. RIPA buffer with 1× protease inhibitor cocktail and 1× phosphatase inhibitor cocktail were used to make whole cell extracts.

For phosphatase assay, 20 μg mouse liver extracts or cell lysis were added 1 μl lambda Protein Phosphatase and diluted to a 30 μL reaction mixture with 3 μl 10 × NEB buffer for Protein Metallophosphatases (PMP) and 3 μl 10 mM MnCl2. The mixtures were then incubated at 30 °C for 45 min.

The protein samples were subjected to denaturing 10% sodium dodecyl sulfate-polyacrylamide gel electrophoresis (SDS-PAGE) and transferred onto polyvinylidene difluoride (PVDF) membranes. Proteins of interest within the membranes were incubated with corresponding antibodies and visualized using the Phototope-HRP Western Blot Detection System (Cell Signaling Technology, USA).

**Tissue collection and antibodies**. Mice were sacrificed at different time points on the first day of constant darkness and liver tissue were weighed and extracted. The fresh liver nuclear extracts were prepared at 4 °C following the protocol of Nuclear Extraction Kit (Active Motif, CA, USA). Additional protein inhibitor cocktail (Roche) and phosphatase inhibitor cocktail (Roche) was added to the nuclear extraction buffer at 2× normal concentration to minimize the loss of unstable proteins.

Immunoblot analysis was conducted following standard protocol with the following antibodies: CK1δ (Abcam, ab85320, 1:1000), PER1 (MBL, PM091, 1:1000), PER2 (MBL, PM083, 1:1000), CLOCK (CST, 5157 S, 1:500), GFP (Sigma-Aldrich, G1544, 1:1000), Flag (Sigma-Aldrich, F1804, 1:1000), HA (Sigma-Aldrich, A2095, 1:1000), Actin (Sigma-Aldrich, A5441, 1:2000) and Tubulin (CST, 5346 s, 1:3000). BMAl1 and CRY1 used in WB (BMAL1 1:1000, CRY1 1:1000), CoIP and ChIP were previously described[84]. PER2pS662 antibody was a gift from Yi Rao's laboratory (Abcam ab206377, 1:1000). Rabbit polyclonal BMAl1 and CRY1 antibodies were generated by Signalway Antibody using synthetic peptides as antigen BMAL1: CSSSILGENPHIGIDMIDNDQGSSSPSNDEA; CRY1: CSQGSGILHYAHGDSQQ THSLKQGRSSAGTG and validated with mouse knockout samples.

**Immunoprecipitation (IP), chromatin immunoprecipitation (ChIP), and ChIP-qPCR**. For immunoprecipitation assay, cell lysates were prepared in 1× RIPA buffer at 4 °C with 2× protease inhibitor cocktail (Roche) and 2× phosphatase inhibitor cocktail (Roche). For ChIP of mouse liver tissues, we used 1% methanol to crosslink DNA to proteins and used a Bioruptor Plus sonication device to fragment chromatin. We used the Active Motif Nuclear Extraction Kit to remove the cytoplasmic protein. ChIP was performed as previously described[84]. Briefly, after sonication, cell debris was removed by centrifugation (14,000 × g, 10 min). The supernatant was preincubated with Dynabeads (Invitrogen) with Rabbit IgG or CRY1 antibody. After incubation overnight in the cold room, magnetic beads were washed from cold low salt wash buffer, high salt wash buffer, LiCl wash buffer to TE buffer. ChIP-ed DNA was eluted, reverse-crosslinked, purified by MinElute PCR Purification Kit (Qiagen), and quantified by Qubit™ dsDNA HS Assay

(Thermo Fisher Scientific) for q-PCR. The q-PCR primers are provided in source data file in Fig. 7b.

**Sequence alignment and protein secondary structure analysis**. Information of protein sequences was searched on National Center for Biotechnology Information (NCBI). The conservation alignment among species was performed by Vector NTI software (Invitrogen). The secondary structure analyses of PERs were performed on the website http://bioinf.cs.ucl.ac.uk/psipred/ and https://alphafold.ebi.ac.uk/entry/O15055.

**E-box activity assay**. HEK293T cells were inoculated in 96-well plates and transfected with vectors for expression of different circadian genes (40 ng of BMAL1, 40 ng of CLOCK or Mutation CLOCK) as well as an E2-box-luciferase reporter plasmid (10 ng of E2 box plasmid) per well[84]. At 24 h after transfection, cells were lysed and examined using a Promega luciferase assay kit following the manufacturer's instructions. The intensities of luciferase activity were quantified with ELISA instrument.

**RNA-sequencing and ChIP-sequencing**. For RNA-sequencing, liver tissue was collected from three mice at each time point. RNA was extracted with TRIzol Reagent (Invitrogen) and quantified with an Agilent 2100 Bioanalyzer. RNA-seq was performed on the DNBSEQ-T7 platform at BGI Genome Center, Shenzhen, China. The expression levels of all NCBI-annotated genes were calculated using RSEM1.2. The genes expression levels were normalized with Z-scores, and visualized in a heatmap generated by using the R package.

ChIP procedures were in line with a standard protocol. Briefly, Douncer-homogenized tissue samples were fixed with 1% formaldehyde for 10 min at room temperature, washed with cold PBS, and then processed to nuclei extraction. Enriched nuclei were re-suspended in 1% SDS lysis buffer for 10 min on ice, and sonicated by Bioruptor (Diagenode) to shear chromatin DNA into 200–500 bp. After sonication, cell debris was removed by centrifugation ($14,000 \times g$, 10 min) and took 10 μl as input. Supernatant was preincubated with Dynabeads (Invitrogen) with IgG for 2 h and then incubated with Dynabeads with BMAl1 antibody. After incubation overnight in the cold room, magnetic beads were washed from cold low salt wash buffer, high salt wash buffer, LiCl wash buffer to TE buffer. ChIP-ed DNA was eluted, reverse-crosslinked, purified by MinElute PCR Purification Kit (Qiagen), and quantified by Qubit™ dsDNA HS Assay (Thermo Fisher Scientific) for sequencing.

**Statistical analysis**. Statistical comparisons between two groups were conducted using Student's *t*-tests. Statistical significance was defined by a *p* value of <0.05. All data are presented as means ± standard deviations of the means (SD). All experimental assays were performed at least twice in independent experiments.

**Reporting summary**. Further information on research design is available in the Nature Research Reporting Summary linked to this article.

## Data availability
RNA-seq and ChIP-seq data that support the findings of this study have been deposited in the NCBI's Sequence Read Archive (SRA) with the with the accession number PRJNA766147 and PRJNA766292. The data supporting the findings of this study are available within the article and its supplementary information files. Source data are provided with this paper.

## Material availability
All materials and mouse lines are available through CAM-SU Genomic Resource Center from the corresponding author upon reasonable request.

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

## Acknowledgements

We thank Drs. Aziz Sancar and Xuemei Cao for discussion of the results, Dr. Xuewu Zhang for assistance with alphafold prediction and Dr. Yi Rao for providing the PER2pS662 antibody. This work was supported by Ministry of Science and Technology 2018YFA0801100 (Y.X.), National Natural Science Foundation of China 31630091(Y.X.), Science and Technology Project of Jiangsu Province BZ2020067 (Y.X.), National Natural Science Foundation of China 31900837 (P.X.), National Institutes of Health R35 GM118118 (Y.L.), Welch Foundation I-1560 (Y.L.), Priority Academic Program Development of the Jiangsu Higher Education Institutes (Y.X.), Lingang Laboratory & National Key Laboratory of Human Factors Engineering Joint Grant (Y.X.), the Special Fund of National Human Genetic Resource Sharing Service Platform YCZYPT[2020]08 (Y.X.), National Center for International Research 2017B01012 (Y.X.).

## Author contributions

Y.X. and Y.L. conceptualized the study, analyzed and interpreted data and wrote the manuscript, which was revised and approved by all authors. Y.A. identified the PER2L730 site and designed and generated the knock-in mice, B.Y. characterized and generated the mice of different genotypes and performed most of the mice experiments, P.X. mapped the region of PER2 interacting with CK1, Y.G. performed the RNA-Seq and ChIP data analyses, Z.L. generated PER2VL729/730G and Per3 mutant models, T.W. and Z.H.L. assisted in mouse works.

## Competing interests

The authors declare no competing interests.

**Additional information**

