## [Peer Review File · Nature Communications]

Decoupling PER phosphorylation, stability and rhythmic expression from circadian clock function by abolishing PER-CK1 interactionREVIEWER COMMENTS

Reviewer #1 (Remarks to the Author):

This comprehensive study by An et al reports novel findings about the critical importance of the CK1-PER physical interactions for the phosphorylation and stability of PER and CLOCK proteins, as well as for driving rhythmic outputs. The authors generated detailed biochemical evidence mapping the PCD domain and used CRISPR to mutate the relevant regions. They also provided in vitro and in vivo data proving the essential role of the PER-CK1 interactions in regulating protein abundance, rhythmicity, locomotor behaviour and rhythmic gene expression in mouse liver. As such, they have revealed a novel dual role of PER/CRY in regulating CLOCK/BMAL1/E-box activities which is conserved from fungi to mammals. Both the biochemical experiments and circadian studies are carefully designed, including proper controls and multiple crossings of animals. The arguments in the paper are overall well supported, for example the authors excluded a role of PER3 by generating triple crosses, and supported a role of CRY as an additional feedback suppression mechanism by conducting a CRY CHIP experiments. Interpretations are cautious but well supported by the data. Discussions are very thoughtful. To my standard, the study is very well carried out with solid findings supporting the authors' claims and represents a significant advance to our understanding of the biochemical basis of the molecular circadian clocks. The manuscript is particularly well written and easy to follow.

There are some relatively minor comments that the authors could consider to further improve the manuscript:

- 1) The title is interesting but somehow does not tell the reader exactly what the paper is about. One suggestion is to be more direct "The CK1-PER interaction is essential for circadian pacemaking properties in mice". Even though mutant mice still show behavioural rhythmicity, they are not at ~24 hour anymore.
- 2) The overall conclusion that the clock is still robust/functional in the absence of CK1/PER docking is not accurate because in peripheral tissues (liver), the low amplitude oscillations of the remaining genes are unlikely to be meaningful in regulating physiology (as shown in Fig.6d). This claim is not well supported unless they can show mice have normal physiology (beyond wheel-running), e.g, glucose metabolism, etc. I agree that the seemingly intact behavioural rhythm is likely due to network properties of the SCN. However, the clock has many other important functions beyond driving locomotion rhythms. Therefore, I would expect the authors to carefully word their conclusions accordingly.
- 3) There was BMAL1 hyperphosphorylation by the PER2 mutant (Fig.4f), but no mentioning of how BMAL1 was hyperphosphorylated, by other kinases or change of a phosphatase?
- 4) One interesting future experiment is to assess how the PER2 mutants generated here will impact on their response to CK1 mutants, e.g., the gain-of-function CK1Tau mutant mice.
- 5) The sampling frequency for liver RNAseq is only one circadian cycle. Although this reviewer appreciates the difficulty of getting sufficient KO animals for a proper circadian time course (2 cycles), the authors should discuss this limitation.
- 6) There are some missing experimental details. For examples, n number/repeats should be included in most of the quantitative experiments. Also, it is unclear how the violin plot in Fig. 6c is generated. There is little detail about the model diagram in Fig. 7b.

Reviewer #2 (Remarks to the Author):

Please see the attached file for comments. –

Review of An et al. "Decoupling PER phosphorylation, stability and rhythmic expression from mammalian circadian clock function" *Nat Comm* 2021

The period, or timing, of mammalian circadian rhythms is highly sensitive to mutations that alter the phosphorylation of PER proteins by CK1, suggesting that this regulatory process plays a critical role in establishing circadian rhythms. Notably, PER proteins bind CK1 with high affinity, such that stable PER-CK1 complexes are found throughout the cell when PER is present over a circadian cycle. This work has the potential to significantly deepen our understanding of PER-CK1 regulation by identifying a residue (L730) in human PER2 that disrupts the stable interaction when mutated. Cell-based and mouse genetic studies of this mutant, with and without knockout of the other *Per* genes, shed new light on the role of CK1 in regulation of PER phosphorylation, stability, nuclear entry, and repressive activity, some of which may be conserved in other eukaryotic clock systems. There are some exciting data here that integrate very well with published studies with more mechanistic data to support a relatively new, multi-stage model for circadian repression. However, there are some instances where the data are of insufficient quality and/or additional data are needed to support the claims being made, especially where they differ from existing models that are supported by the breadth of literature in the field. Furthermore, there are a number of missed opportunities to attribute prior foundational work through the omission or incomplete attribution of citations. Altogether, these factors diminish the stated novelty of some of the claims, potentially limiting the rigor and impact of this study on the field. Many of these issues could be addressed by providing a more thorough framework of this work in the context of the literature and attenuating claims that lack direct support from the data. If these concerns are addressed, this work could still have a large impact on the field by providing genetic model systems that dissect CK1-dependent and independent modes of repression in the mammalian circadian clock.

Major:

1. From the outset, the stated novelty of some of this work is overshadowed by the omission of and/or insufficient attribution of key data from citations. Some data were published as long as 16 years ago, while others were published just 6-12 months ago, but certainly long enough ago to be properly attributed. I do not think that establishing these prior precedents detracts in any way from the work here.
 - Fig. 1a-c duplicate prior results from >15 years ago without proper attribution. Lee, C. et al. 2004 *Mol Cell Biol* defined the boundaries of the Casein Kinase Binding Domain (CKBD) and demonstrated its importance in scaffolding CK1 and the phosphorylation of PER by CK1. Virshup and colleagues (Eide et al. 2005 *Mol Cell Biol*) identified two motifs in the CKBD of PER2, one of which encompasses the region studied here, as being essential for CK1-mediated phosphorylation of PER2 and its turnover by β -TrCP. This discovery, which apparently laid the foundation for this work, should be made clear when this paper is cited instead of just citing it in the context of CK1-dependent regulation by β -TrCP.
 - Vis-à-vis the argument about evolutionary conservation, no reference is made to two studies of a highly conserved region in *Drosophila* PER and its role on CK1 binding and PER repressive activity, as shown by Nawathean, P. et al. 2007 *Mol Cell Biol* and Kim. E. et al. 2007 *Mol Cell Biol*.
 - The work by Cao et al. 2021 *PNAS* describing a role for CRY and PER-dependent recruitment of CK1 in 'displacement-type repression' of CLOCK/BMAL1 should be properly cited as literature precedent it was published almost a year ago in Jan 2021, and not added after the fact as a sort of 'note in proof'. Earlier work by Weitz and colleagues (Aryal, R. et al. 2017 *Mol Cell*) is also not cited in the context of first

showing CRY and PER-dependent CLOCK phosphorylation using complexes isolated from native liver samples *in vivo*.

- The statement that “PER proteins function as the main negative elements in the mammalian circadian negative feedback loop” neglects to consider data that repressive function by PER requires CRYs (Chiou, Y. et al. 2016 *PNAS* and Cao, Y. et al. 2021 *PNAS*). Moreover, the conclusion that circadian rhythms can be found in SCN of some neonatal *Cry1/2* knockouts, and therefore that *Crys* are not important for circadian repression, relies on the rather niche point that the intercellular coupling in the SCN can compensate for low amplitude cellular rhythms—this misrepresents a strong body of literature that *Crys* are in fact important for robust circadian rhythms at the cellular level, including in dissociated SCN neurons (Liu, A. et al. 2007 *Cell*, Evans, J. et al. 2012 *J Biol Rhythms* and many others) where they act as direct repressors of CLOCK/BMAL1 activity as well as recruiting PER-CK1 complexes to the transcription factor. In my opinion, it does not diminish from the importance of this work to provide an accurate statement on the role of *Crys* in the mammalian circadian clock mechanism.
 - The characterization that PERs obligately regulate CRY1 nuclear entry relies on older data based primarily on overexpression assays, which has been called into question by newer data looking at endogenously driven proteins in cycling cells (Gabriel, C. et al. 2021 *Nat Comm*).
2. The statement that PER stability appears to correlate with period length is true, but the statement that “PER in cells with short period length has reduced turnover rate” is incorrect—the opposite is true. Moreover, citations #22 and 23 made in support of this statement have nothing to do with mammalian PER and mutants that affect period, but with proteins in the *Neurospora* and *Drosophila* clocks. There are a number of appropriate citations to support the corrected statement, including from one of the senior authors’ own work.
 3. The conclusion that PER1 and PER2 stably bind CK1 while PER3 does not isn’t well supported by the data in Fig. 1a, where PER1 and 3 both appear to bind weakly, although both are also expressed to a lower level than PER2 in the input. Also, the co-IPs that re-mapped the CKBD (after it was already defined by Lee at al., see above) in Fig. 1b are not rigorous, as the expression of CK1 in several samples where binding is not seen is quite low. The same is true for Fig. 1f, as CK1 expression in the L730G mutant is considerably lower than the other samples. CK1 should be expressed similarly across the samples to make even a qualitative conclusion about the ‘strength’ of binding in this non-equilibrium binding assays.
 4. The explanation for potential differences between *in vitro* and *in vivo* phosphorylation data could be framed more holistically. Aside from citation #37, which used an engineered kinase substrate based on PER2 (i.e., CK1tide), all of the other cited studies focused on physiologically relevant sites identified *in vivo* and used gold-standard kinetic assays with substrate titrations to characterize kinase activity. Notably, the K_m s of the kinase for native PER2 substrates in the “unanchored” state are quite high, so it is reasonable that forming a stoichiometric complex could have a dramatic effect on kinase activity under cellular conditions by enhancing enzyme efficiency through enforced proximity. Portraying these differences as a cautionary “contrast” between *in vitro* and *in vivo* data introduces unnecessary drama that is really just based on the known effect of substrate concentration on enzyme activity, a basic principle of enzyme kinetics.
 5. The authors are commended for trying to identify specific CLOCK phosphorylation sites in Fig. 3c, but unfortunately, the western blot data are of too low quality to be convincing. Moreover, the statement that the M2 and M3 mutants activate CLOCK/BMAL1 in Fig. 3d is unsubstantiated because they are expressed to a much higher level than WT CLOCK

in Fig. 3c. Given these differences in expression, western blot data should be included of the samples used in the luciferase assays.

6. The western blot of nuclear extracts in Fig. 4h is quite interesting given the longstanding assumption that CK1 somehow regulates PER2 nuclear entry, but it would be better to see a proper subcellular fractionation (cyto and nuclear) of liver samples or use a cytoplasmic marker to demonstrate a clean nuclear fractionation. If the fraction does represent the nuclear proteins, why do the authors not make the powerful conclusion that CK1 phosphorylation is apparently not required for nuclear entry? This seems pretty compelling to me and is perhaps the first time that this is unequivocally addressed in the literature.
7. In Fig. 6a-b, does it make sense to identify rhythmic genes in MetaCycle using a period of 24 h if the mutant has a significantly different period? Are there other ways to analyze the data that could circumvent this assumption? Some of the data in the apparently 'arrhythmic' cluster look quite rhythmic, although certainly of lower amplitude.
8. There are no data to support the "opposing roles" of PER in regulating CLOCK:BMAL1 activity, which is based on extensive speculation that unphosphorylated PER2 is found with CLOCK/BMAL1 and CRY1 on DNA to assist CRY1-mediated repression as depicted in the cartoon in Fig. 7b. ChIP-Seq data from Koike, N. et al. 2012, *Science* showed clock gene occupancy in the liver over a circadian cycle, supporting a PER-independent role for CRY1 repression from CT0-4, so it's not clear why the authors only looked at CRY1 binding by ChIP and only at CT8 and CT16 in Fig. 7. This model lacks experimental support for its fundamental conclusions.
9. There is virtually no discussion of the long period in the *Per2^{m/m}* mutant and why this period is long while the *Per1^{-/-}Per2^{m/m}* mutant has a short period. The statement in the Discussion that "the long period rhythms of the *Per2^{m/m}* mice indicate that this feedback loop plays an important role in period determination" is circular and doesn't explain anything. This needs to be addressed in some way.

Minor:

1. The statement "CK1 forms stable complexes with clock proteins" in the abstract could be clarified—based on published literature and this manuscript, CK1 only forms a stable and direct interaction with PER1 and PER2 proteins, which then form large complexes with other clock proteins.
2. Citation #22 is used incorrectly in support of the phosphoswitch model of PER2 degradation. See also Major point #2—I did not go through the whole manuscript to verify the suitability of the other references, but it might be warranted.
3. The lanes for the actin loading control for the CHX timecourse in Fig. 2c are overloaded.
4. The cartoon in Fig. 2f should attempt to illustrate trans-phosphorylation by PER1 in the *Per2* mutant. As it currently stands, it is not particularly helpful in illustrating that point.
5. The cartoon in Fig. 3e should be accurate in depicting CRY as binding to both CLOCK and BMAL1 (Xu, H. et al. 2015 *NSMB*) and being required for PER-CK1 recruitment (Chiou, Y. et al. 2016 *PNAS*). There is not any strong evidence that PERs bind directly to CLOCK in the literature.
6. There was a note left in the legend for Fig. 3 to add more detail to the methods.
7. The statement "We found that PER acts a scaffold for CK1 to promote phosphorylation of different PER proteins..." in the Discussion could be made clearer.
8. If space is not limited, details about BMAL1 and CRY1 antibodies should be provided in the methods and not just referred to with a citation. The PER2 pS662 antibody from Yi Rao's lab does not have a citation in support of it; please provide one or information on how the antibody was made and purified.

Reviewer #3 (Remarks to the Author):

In this article, the authors identify a region in PER that is critical for its interaction with CK1. In vivo, PER mutant (unable to bind CK1) animals had robust short period locomotor activity and low amplitude rhythms. Furthermore, PER mutation abolishes PER phosphorylation and CLOCK hyperphosphorylation resulting in PER stability. The author present results that favor a model in which the circadian clock can function independently of PER phosphorylation and abundance and that period length can be uncoupled from PER stability.

Major comments:

a- The authors should either delete the statement that reads "In addition, the hPER3-CK1d association was weak." or provide biochemical evidence in the form of binding constant values to state the above.

b- The authors state that "amino acids 729-738 of hPER2 are required for hPER2-CK1d association" and that "this region is conserved" among PER proteins and that "this region is predicted to form an alpha helix in all three proteins." The authors state the prediction was carried out using alphaFold. Please, upload the structure prediction and validation results from alphaFold as supplementary material.

c- Then, the authors mutate residues V729 and L730 to Gly and show that mutation in these two residues abrogate PER2 binding to CK1d. However, the choice of Gly to substitute residues 729 and 730 in an alpha helix is unfortunate. Glycine and proline residues are rarely found in alpha helices because they tend to destabilize the α -helix due to the conformational freedom of Gly that favors the unfolding of the helix (the opposite with proline that makes it more rigid). Anyway, the point is that by mutating these two residues to Gly, the helix will be destabilized and unlikely to bind CK1. This result might mislead the authors to believe that those two residues are key to the interaction when, in fact, the helix is unfolded. If the authors want to claim these two residues are specifically relevant to the interaction, they should perform a similar IP experiment replacing V729 and L730 by Alanine to begin with.

d- Fig. 2a: lane 5 (counting from the left) why is the amount of Per2 lower than in lane 4. If the deletion prevents binding of CK1 and phosphorylation, the levels should be comparable. Unless, the deletion of the helix favors PER2 instability and degradation. In favor of this hypothesis is the result shown using the GG mutant in the same figure (lanes 7 and 8). The authors do not provide any explanation for the obvious differences in protein levels for PER2 shown among lanes. The authors need to show the stability of PER2 and PER delta(729-738) and PER2 GG are comparable. Otherwise, there is a risk that the conclusions turn out to be a technical artifact.

e- Why didn't the authors perform the experiment in Fig. 2C using the Delta729-738 form? This reviewer would like to see a comparison between the half-life of PER2 and the delta form when co-transfected with CK1.

f- Fig. 3a: the problem with this panel is that the authors transfect cells with HA-CLOCK and then use an anti-CLOCK antibody to see the phosphorylation shift instead of an HA antibody. The doublet could be the authors see the endogenous CLOCK and the HA-Clock (few amino acids difference) with the same antibody. In addition, this reviewer would like to see the experiment transfecting either PER2 or PER2 L730G and detecting the endogenous proteins and shifts. The system, as constructed, is too artificial (at least 5 co-transfections simultaneously, what is the point of doing in cells?).

g- Fig. 3b: a system with an inducible promoter would have been a more elegant experiment.

h- Fig. 3c: how do the authors explain that lanes 2 and 3 show the same level of CLOCK hyperphosphorylation but only one overexpresses PER2?

i- Fig. 4c needs to be biochemically validated by the experiments requested above. Fig. 5 shows mutant animals have a strong phenotype but, again, biochem experiments are needed to rule out any artifact.

Minor comments:

a- Please revise the following sentence (2nd paragraph, Results section) and remove the second "association" word in the first sentence.

b- Fig. 3c: BMAL is misspelled (reads BAML1 in the figure)

Reviewer #1

This comprehensive study by An et al reports novel findings about the critical importance of the CK1-PER physical interactions for the phosphorylation and stability of PER and CLOCK proteins, as well as for driving rhythmic outputs. The authors generated detailed biochemical evidence mapping the PCD domain and used CRISPR to mutate the relevant regions. They also provided in vitro and in vivo data proving the essential role of the PER-CK1 interactions in regulating protein abundance, rhythmicity, locomotor behaviour and rhythmic gene expression in mouse liver. As such, they have revealed a novel dual role of PER/CRY in regulating CLOCK/BMAL1/E-box activities which is conserved from fungi to mammals. Both the biochemical experiments and circadian studies are carefully designed, including proper controls and multiple crossings of animals. The arguments in the paper are overall well supported, for example the authors excluded a role of PER3 by generating triple crosses, and supported a role of CRY as an additional feedback suppression mechanism by conducting a CRY CHIP experiments. Interpretations are cautious but well supported by the data. Discussions are very thoughtful. To my standard, the study is very well carried out with solid findings supporting the authors' claims and represents a significant advance to our understanding of the biochemical basis of the molecular circadian clocks. The manuscript is particularly well written and easy to follow.

1. The title is interesting but somehow does not tell the reader exactly what the paper is about. One suggestion is to be more direct "The CK1-PER interaction is essential for circadian pacemaking properties in mice". Even though mutant mice still show behavioural rhythmicity, they are not at ~24 hour anymore.

Response: Although the role of PER-CK1 interaction is important for period determination, we feel that maintenance of circadian rhythmicity despite of the loss of PER phosphorylation, PER rhythmic expression and a key negative feedback process in the mutant mice is the most important conclusion of this study. In the revised paper, we also added data showing that additional circadian rhythms of physiological processes (oxygen consumption rate (VO₂), respiratory exchange ratio (RER) and activity counts) are also maintained in the mutant mice. We have now changed to the title of the paper to "Decoupling PER phosphorylation, stability and rhythmic expression from circadian clock function by abolishing PER-CK1 interaction".

2. The overall conclusion that the clock is still robust/functional in the absence of CK1/PER docking is not accurate because in peripheral tissues (liver), the low amplitude oscillations of the remaining genes are unlikely to be meaningful in regulating physiology (as shown in Fig.6d). This claim is not well supported unless they can show mice have normal physiology (beyond wheel-running), e.g, glucose metabolism, etc. I agree that the seemingly intact behavioural rhythm is likely due to network properties of the SCN. However, the clock has many other important functions beyond driving locomotion rhythms. Therefore, I would expect the authors to carefully word their conclusions accordingly.

Response: Thanks for the suggestion. To address this concern, we compared circadian rhythms of metabolism and activity of the wild-type and mutant mice in metabolic cages. As shown in the revised Figure 5c, the circadian rhythms of oxygen consumption rate (VO₂), respiratory exchange ratio (RER) and activity counts were comparable in the wild-type and *Per1*^{-/-};*Per2*^{m/m} mice. Together, these results suggest that despite the low amplitude oscillations of clock genes in the mutant mice, robust circadian rhythms of physiological processes can be maintained.

3. There was BMAL1 hyperphosphorylation by the PER2 mutant (Fig.4f), but no mentioning of how BMAL1 was hyperphosphorylated, by other kinases or change of a phosphatase?

Response: Thanks for the suggestion. How BMAL1 becomes hyperphosphorylated is currently not known. We have now added discussion on BMAL1 phosphorylation and its regulation in the discussion. It was previously shown that BMAL1 is phosphorylated by Casein kinase (CK)-2 α and CRY mediates periodic binding of CK2 β to BMAL1 to inhibit BMAL1 phosphorylation by CK2 α ^{72,73}. We propose that PER and PER-CK1 interaction are also required for the inhibitory role of CRY in BMAL1 phosphorylation.

4. One interesting future experiment is to assess how the PER2 mutants generated here will impact on their response to CK1 mutants, e.g., the gain-of-function CK1Tau mutant mice.

Response: This is a great suggestion. We hope we will be able to do such an experiment in the future if we can generate/obtain the CK1 tau mutant mice.

5. The sampling frequency for liver RNAseq is only one circadian cycle. Although this reviewer appreciates the difficulty of getting sufficient KO animals for a proper circadian time course (2 cycles), the authors should discuss this limitation.

Response: Due to the cost and the need for large number of mutant mice in this experiment, we only collected RNA-seq samples for one circadian cycle. However, we had 3 mice for each point. In addition, our comparison of the WT and mutant mice showed dramatic difference in rhythmic gene expression profiles. As suggested, we now mentioned this limitation in the text.

6. There are some missing experimental details. For examples, n number/repeats should be included in most of the quantitative experiments. Also, it is unclear how the violin plot in Fig. 6c is generated. There is little detail about the model diagram in Fig. 7b.

Response: Thanks for the suggestions. As suggested, we have now added the requested information in the figure legends. Detailed description of the model figure in Figure 7c is also added in the Figure 7 legends.

Reviewer #2

The period, or timing, of mammalian circadian rhythms is highly sensitive to mutations that alter the phosphorylation of PER proteins by CK1, suggesting that this regulatory process plays a critical role in establishing circadian rhythms. Notably, PER proteins bind CK1 with high affinity, such that stable PER-CK1 complexes are found throughout the cell when PER is present over a circadian cycle. This work has the potential to significantly deepen our understanding of PERCK1 regulation by identifying a residue (L730) in human PER2 that disrupts the stable interaction when mutated. Cell-based and mouse genetic studies of this mutant, with and without knockout of the other *Per* genes, shed new light on the role of CK1 in regulation of PER phosphorylation, stability, nuclear entry, and repressive activity, some of which may be conserved in other eukaryotic clock systems. There are some exciting data here that integrate very well with published studies with more mechanistic data to support a relatively new, multistage model for circadian repression. However, there are some instances where the data are of insufficient quality and/or additional data are needed to support the claims being made, especially where they differ from existing models that are supported by the breadth of literature in the field. Furthermore, there are a number of missed opportunities to attribute prior foundational work through the omission or incomplete attribution of citations. Altogether, these factors diminish the stated novelty of some of the claims, potentially limiting the rigor and impact of this study on the field. Many of these issues could be addressed by providing a more thorough framework of this work in the context of the literature and attenuating claims that lack direct support from the data. If these concerns are addressed, this work could still have a large impact on the field by providing genetic model systems that dissect CK1-dependent and independent modes of repression in the mammalian circadian clock.

Response: We really appreciate the truly insightful comments and constructive suggestions of this reviewer. By revising our paper based these suggestions, we believe our paper is now greatly strengthened.

Major:

1. From the outset, the stated novelty of some of this work is overshadowed by the omission of and/or insufficient attribution of key data from citations. Some data were published as long as 16 years ago, while others were published just 6-12 months ago, but certainly long enough ago to be properly attributed. I do not think that establishing these prior precedents detracts in any way from the work here.

Fig. 1a-c duplicate prior results from >15 years ago without proper attribution. Lee, C. et al. 2004 *Mol Cell Biol* defined the boundaries of the Casein Kinase Binding Domain (CKBD) and demonstrated its importance in scaffolding CK1 and the phosphorylation of PER by CK1. Virshup and colleagues (Eide et al. 2005 *Mol Cell Biol*) identified two motifs in the CKBD of PER2, one of which encompasses the region studied here, as being essential for CK1-mediated phosphorylation of PER2 and its turnover by β -TrCP. This discovery, which apparently laid the foundation for

this work, should be made clear when this paper is cited instead of just citing it in the context of CK1-dependent regulation by β -TrCP.

Response: Thanks for the suggestions. We have now incorporated these information and references before we described our results.

Vis-à-vis the argument about evolutionary conservation, no reference is made to two studies of a highly conserved region in *Drosophila* PER and its role on CK1 binding and PER repressive activity, as shown by Nawathean, P. et al. 2007 *Mol Cell Biol* and Kim. E. et al. 2007 *Mol Cell Biol*.

Response: Thanks for the suggestions. We have now revised the introduction and results to include these information and references.

The work by Cao et al. 2021 *PNAS* describing a role for CRY and PER-dependent recruitment of CK1 in ‘displacement-type repression’ of CLOCK/BMAL1 should be properly cited as literature precedent it was published almost a year ago in Jan 2021, and not added after the fact as a sort of ‘note in proof’. Earlier work by Weitz and colleagues (Aryal, R. et al. 2017 *Mol Cell*) is also not cited in the context of first showing CRY and PER-dependent CLOCK phosphorylation using complexes isolated from native liver samples *in vivo*.

Response: Thanks for the suggestions. We have revised the introduction and results by including these information and references.

1. The statement that “PER proteins function as the main negative elements in the mammalian circadian negative feedback loop” neglects to consider data that repressive function by PER requires CRYs (Chiou, Y. et al. 2016 *PNAS* and Cao. Y. et al. 2021 *PNAS*). Moreover, the conclusion that circadian rhythms can be found in SCN of some neonatal *Cry1/2* knockouts, and therefore that *Crys* are not important for circadian repression, relies on the rather niche point that the intercellular coupling in the SCN can compensate for low amplitude cellular rhythms—this misrepresents a strong body of literature that *Crys* are in fact important for robust circadian rhythms at the cellular level, including in dissociated SCN neurons (Liu, A. et al. 2007 *Cell*, Evans, J. et al. 2012 *J Biol Rhythms* and many others) where they act as direct repressors of CLOCK/BMAL1 activity as well as recruiting PER-CK1 complexes to the transcription factor. In my opinion, it does not diminish from the importance of this work to provide an accurate statement on the role of *Crys* in the mammalian circadian clock mechanism.

Response: Thanks for the suggestions. We have now rewritten part of the introduction of the paper to include these information and references.

The characterization that PERs obligately regulate CRY1 nuclear entry relies on

older data based primarily on overexpression assays, which has been called into question by newer data looking at endogenously driven proteins in cycling cells (Gabriel, C. et al. 2021 *Nat Comm*).

Response: Thanks for the suggestions. We have now delete this statement in the paper.

2. The statement that PER stability appears to correlate with period length is true, but the statement that “PER in cells with short period length has reduced turnover rate” is incorrect—the opposite is true. Moreover, citations #22 and 23 made in support of this statement have nothing to do with mammalian PER and mutants that affect period, but with proteins in the *Neurospora* and *Drosophila* clocks. There are a number of appropriate citations to support the corrected statement, including from one of the senior authors’ own work.

Response: Thanks for pointing out this mistake. We have now corrected this mistake and replaced these references with new references.

3. The conclusion that PER1 and PER2 stably bind CK1 while PER3 does not isn’t well supported by the data in Fig. 1a, where PER1 and 3 both appear to bind weakly, although both are also expressed to a lower level than PER2 in the input. Also, the coIPs that re-mapped the CKBD (after it was already defined by Lee at al., see above) in Fig. 1b are not rigorous, as the expression of CK1 in several samples where binding is not seen is quite low. The same is true for Fig. 1f, as CK1 expression in the L730G mutant is considerably lower than the other samples. CK1 should be expressed similarly across the samples to make even a qualitative conclusion about the ‘strength’ of binding in this non-equilibrium binding assays.

Response: Thanks for the suggestions. The reviewer is right that the result in Figure 1a is not quantitative enough to make a firm conclusion. In the revised paper, we removed the statement of “PER3 binds weakly to CK1”.

In our experiments, we consistently observed that when CK1 was expressed alone or co-expressed with hPER2 mutant that cannot interact with CK1, CK1 levels were low. This was the reason for the reduced CK1 levels in the lane 2, 4 and 6 compared to lane 3 and 5 in Figure 1f. Similarly, the input CK1 levels were also low in lane 1, 5 and 9 of Figure 1b. These observations suggest that PER-CK1 interaction can stabilize CK1 when they co-expressed in cells. We have now added this observation in the text. We agree with this reviewer that a quantitative conclusion on the binding strength could not be determined when CK1 expression level was not even. We have repeated this experiment many times and could not observe CK1 co-precipitated with PERL730G and PER2 VL729-730GG even under longer exposure. As a result, we are very confident with our conclusion.

4. The explanation for potential differences between in vitro and in vivo phosphorylation data could be framed more holistically. Aside from citation #37, which

used an engineered kinase substrate based on PER2 (i.e., CK1tide), all of the other cited studies focused on physiologically relevant sites identified in vivo and used gold-standard kinetic assays with substrate titrations to characterize kinase activity. Notably, the K_m s of the kinase for native PER2 substrates in the "unanchored" state are quite high, so it is reasonable that forming a stoichiometric complex could have a dramatic effect on kinase activity under cellular conditions by enhancing enzyme efficiency through enforced proximity. Portraying these differences as a cautionary "contrast" between in vitro and in vivo data introduces unnecessary drama that is really just based on the known effect of substrate concentration on enzyme activity, a basic principle of enzyme kinetics.

Response: We appreciate the suggestion. As suggested, we now rewrote these sentences in the introduction and removed the "contrast" description in the results.

5. The authors are commended for trying to identify specific CLOCK phosphorylation sites in Fig. 3c, but unfortunately, the western blot data are of too low quality to be convincing. Moreover, the statement that the M2 and M3 mutants activate CLOCK/BMAL1 in Fig. 3d is unsubstantiated because they are expressed to a much higher level than WT CLOCK in Fig. 3c. Given these differences in expression, western blot data should be included of the samples used in the luciferase assays.

Response: To address these concerns, we have now redone the experiments in the original Figure 3c and 3d (Revised Fig. 3d & e). As suggested, we now added the western blot data for 3d (revised Fig. 3e). We believe the revised data is now of high quality.

6. The western blot of nuclear extracts in Fig. 4h is quite interesting given the longstanding assumption that CK1 somehow regulates PER2 nuclear entry, but it would be better to see a proper subcellular fractionation (cyto and nuclear) of liver samples or use a cytoplasmic marker to demonstrate a clean nuclear fractionation. If the fraction does represent the nuclear proteins, why do the authors not make the powerful conclusion that CK1 phosphorylation is apparently not required for nuclear entry? This seems pretty compelling to me and is perhaps the first time that this is unequivocally addressed in the literature.

Response: As suggested, we have now performed western blot analyses of two control proteins: TUBULIN (cytoplasmic control) and LAMINB1 (nuclear control). As shown in Figure S4, α -TUBULIN was not detected in the nuclear samples and LAMINB1 was enriched in the nuclear samples, indicating that our nuclear samples were free of cytoplasmic proteins.

Thanks to the suggestion of this reviewer, we now added a statement in the text stating that CK1 phosphorylation is not required for PER nuclear entry.

7. In Fig. 6a-b, does it make sense to identify rhythmic genes in MetaCycle using a

period of 24 h if the mutant has a significantly different period? Are there other ways to analyze the data that could circumvent this assumption? Some of the data in the apparently ‘arrhythmic’ cluster look quite rhythmic, although certainly of lower amplitude.

Response: As suggested, we compared the results of the MetaCycle analyses using a period of 24 and 22.3 h (period of the *Per1*^{-/-};*Per2*^{m/m} mice) and found that 95% of the identified rhythmic genes in the 24 h period analysis were also found to be rhythmic genes when a period of 22.3 was used. Thus, the results of these analyses are highly similar each other and the use of 24 h in the analysis should not change the conclusion. We have now added a statement about this in the text.

8. There are no data to support the "opposing roles" of PER in regulating CLOCK:BMAL1 activity, which is based on extensive speculation that unphosphorylated PER2 is found with CLOCK/BMAL1 and CRY1 on DNA to assist CRY1-mediated repression as depicted in the cartoon in Fig. 7b. ChIP-Seq data from Koike, N. et al. 2012, *Science* showed clock gene occupancy in the liver over a circadian cycle, supporting a PER-independent role for CRY1 repression from CT0-4, so it's not clear why the authors only looked at CRY1 binding by ChIP and only at CT8 and CT16 in Fig. 7. This model lacks experimental support for its fundamental conclusions.

Response: Thanks for raising this issue. The poor writing of the original draft on this section created some confusion on this issue. The “opposing roles” of PER should mean the two opposite roles of PER-CK1 interaction on CLOCK-BMAL1 activity: promoting CLOCK-BMAL1 removal from E-boxes but reducing the inhibition of CLOCK-BMAL1 activity on DNA. The latter is indicated by the constant low levels of *Dbp*, *Nr1d1*, *Cry1* mRNAs (Fig. 6d) despite the increased BMAL1 levels at E-boxes in the *Per1*^{-/-};*Per2*^{m/m} mice. The transcription of these genes is mostly driven by the CLOCK-BMAL1 transcription activation activity and is repressed by PER expression.

As suggested, we performed CRY1 ChIP assays at different time points (revised Fig 7b). In addition, we added mPER2 IP result (revised Fig. 7a) showing that PER-CRY1 association is constantly high in the mutant mice despite the reduced CRY1 protein levels.

We have now completely re-written this section of results and changed its sub-title to “PER-CK1 interaction reduces inhibition of CLOCK-BMAL activity on DNA”.

9. There is virtually no discussion of the long period in the *Per2*^{m/m} mutant and why this period is long while the *Per1*^{-/-};*Per2*^{m/m} mutant has a short period. The statement in the Discussion that “the long period rhythms of the *Per2*^{m/m} mice indicate that this feedback loop plays an important role in period determination” is circular and doesn't explain anything. This needs to be addressed in some way.

Response: We think the strong inhibition of the CLOCK-BMAL1 E-box activity in the

Per1^{-/-};*Per2*^{m/m} mice may explain its short period phenotype (see above). In the discussion, we stated that “In the absence of the PER-CK1 association, the PER-CRY-dependent mechanism results in an increased inhibition of CLOCK-BMAL1 activity, which may explain the short period phenotype in the *Per1*^{-/-};*Per2*^{m/m} mice.” We also added a similar statement in the section describing the results under “PER-CK1 interaction reduces inhibition of CLOCK-BMAL activity on DNA”.

Minor:

1. The statement “CK1 forms stable complexes with clock proteins” in the abstract could be clarified—based on published literature and this manuscript, CK1 only forms a stable and direct interaction with PER1 and PER2 proteins, which then form large complexes with other clock proteins

Response: Revised as suggested.

2. Citation #22 is used incorrectly in support of the phosphoswitch model of PER2 degradation. See also Major point #2—I did not go through the whole manuscript to verify the suitability of the other references, but it might be warranted.

Response: Thanks for pointing this out. This reference has been replaced with other references. We have also checked the entire manuscript for accuracy as suggested.

3. The lanes for the actin loading control for the CHX timecourse in Fig. 2c are overloaded.

Response: We have replaced it with a shorter exposure. The protein was evenly loaded in different samples.

4. The cartoon in Fig. 2f should attempt to illustrate trans-phosphorylation by PER1 in the *Per2* mutant. As it currently stands, it is not particularly helpful in illustrating that point.

Response: We have revised the illustration to make it clearer.

5. The cartoon in Fig. 3e should be accurate in depicting CRY as binding to both CLOCK and BMAL1 (Xu, H. et al. 2015 *NSMB*) and being required for PER-CK1 recruitment (Chiou, Y. et al. 2016 *PNAS*). There is not any strong evidence that PERs bind directly to CLOCK in the literature.

Response: We have revised the illustration as suggested. Fig.3e is now the revised Fig. 3b.

6. There was a note left in the legend for Fig. 3 to add more detail to the methods.

Response: Thank you! It has been corrected.

7. The statement “We found that PER acts a scaffold for CK1 to promote phosphorylation of different PER proteins...” in the Discussion could be made clearer.

Response: We revised this sentence.

8. If space is not limited, details about BMAL1 and CRY1 antibodies should be provided in the methods and not just referred to with a citation. The PER2 pS662 antibody from Yi Rao’s lab does not have a citation in support of it; please provide one or information on how the antibody was made and purified.

Response: The antibody information is now added to the methods.

Reviewer #3

In this article, the authors identify a region in PER that is critical for its interaction with CK1. In vivo, PER mutant (unable to bind CK1) animals had robust short period locomotor activity and low amplitude rhythms. Furthermore, PER mutation abolishes PER phosphorylation and CLOCK hyperphosphorylation resulting in PER stability. The author present results that favor a model in which the circadian clock can function independently of PER phosphorylation and abundance and that period length can be uncoupled from PER stability.

1- The authors should either delete the statement that reads “In addition, the hPER3-CK1d association was weak.” or provide biochemical evidence in the form of binding constant values to state the above.

Response: We have now deleted this statement as suggested.

2- The authors state that “amino acids 729-738 of hPER2 are required for hPER2-CK1d association” and that “this region is conserved” among PER proteins and that “this region is predicted to form an alpha helix in all three proteins.” The authors state the prediction was carried out using alphafold. Please, upload the structure prediction and validation results from alphafold as supplementary material.

Response: As suggested, we have added the alphafold structure predictions of this region with and without the mutations in Figure 1f. As can be seen from the result, the GG mutations did not alter the helical structure of this domain.

3- Then, the authors mutate residues V729 and L730 to Gly and show that mutation in these two residues abrogate PER2 binding to CK1d. However, the choice of Gly to substitute residues 729 and 730 in an alpha helix is unfortunate. Glycine and proline residues are rarely found in alpha helices because they tend to destabilize the α -helix due to the conformational freedom of Gly that favors the unfolding of the helix (the opposite with proline that makes it more rigid). Anyway, the point is that by mutating these two residues to Gly, the helix will be destabilized and unlikely to bind CK1. This result might mislead the authors to believe that those two residues are key to the interaction when, in fact, the helix is unfolded. If the authors want to claim these two residues are specifically relevant to the interaction, they should perform a similar IP experiment replacing V729 and L730 by Alanine to begin with.

Response: The mutations to glycine residues was used after discussions with a couple of structural experts at UT Southwestern, including one HHMI investigator next to our lab. Although Glycine residues are not favored in α -helices, they are found in natural α -helices and do not necessarily disrupt α -helix formation. To address this concern, we have now performed alphafold structure predictions after the Gly mutations. As shown in Fig. 1f, the predicted overall α -helical structures of the WT sequence and its Gly mutation are almost identical. Therefore, the lack of PER-CK1 interaction after the mutation is not due to the disruption of the α -helical structure. We have now added a statement in the text on this issue.

4- Fig. 2a: lane 5 (counting from the left) why is the amount of Per2 lower than in lane 4. If the deletion prevents binding of CK1 and phosphorylation, the levels should be comparable. Unless, the deletion of the helix favors PER2 instability and degradation. In favor of this hypothesis is the result shown using the GG mutant in the same figure (lanes 7 and 8). The authors do not provide any explanation for the obvious differences in protein levels for PER2 shown among lanes. The authors need to show the stability of PER2 and PER2 Δ (729-738) and PER2 GG are comparable. Otherwise, there is a risk that the conclusions turn out to be a technical artifact. Why didn't the authors perform the experiment in Fig. 2C using the Δ 729-738 form? This reviewer would like to see a comparison between the half-life of PER2 and the Δ form when co-transfected with CK1.

Response: There are some variations of protein expression levels in transfection experiments and low PER level in lane 5 of Figure 2a was not seen in other independent experiments. Below is a result from an independent experiment showing that the 729-738 deletion and PER2 GG did not have low protein levels. We have now added a statement on this in the Figure legend.

As suggested, we compared the PER stability of Delta729-738, PER2 GG and PER2730G. The results showed that there were no statistical differences of their decay rates (see below) and all were much more stable than the WT PER2.

6- Fig. 3a: the problem with this panel is that the authors transfect cells with HA-CLOCK and then use an anti-CLOCK antibody to see the phosphorylation shift instead of an HA antibody. The doublet could be the authors see the endogenous CLOCK and the HA-Clock (few amino acids difference) with the same antibody. In addition, this reviewer would like to see the experiment transfecting either PER2 or PER2 L730G and detecting the endogenous proteins and shifts. The system, as constructed, is too artificial (at least 5 co-transfections simultaneously, what is the point of doing in cells?).

Response: Note that Lane 1 in 3a used 293 cells without plasmid transfection and was served as the negative control. It is clear that our CLOCK antibody was not sensitive enough to detect the endogenous CLOCK protein under this condition. Thus, the signals detected in cells that were transfected with HA-CLOCK are all HA-CLOCK proteins. In addition, we also performed western blot using HA antibody (attached result below). The HA-CLOCK signals are the same as we observed in Figure 3a. We now added a statement on this in the figure legend.

We agree that all transient transfection experiments are somewhat artificial, which is the reason we generated the knock-in mPER2L730G mice. It is very important to note that our results on PER and CLOCK phosphorylation were all confirmed in vivo using the *Per1*^{-/-};*Per2*^{m/m} mice (Fig. 4).

7- Fig. 3b: a system with an inducible promoter would have been a more elegant experiment.

Response: We agree that an inducible promoter will be a better experiment. But it is very important to note that our results on PER and CLOCK phosphorylation were all confirmed in vivo using the knock-in *Per1*^{-/-};*Per2*^{m/m} mice (Fig. 4).

8- Fig. 3c: how do the authors explain that lanes 2 and 3 show the same level of CLOCK hyperphosphorylation but only one overexpresses PER2?

Response: The quality of the original Fig. 3c was not the best which might have caused this confusion. We have now repeated this experiment and the current results are much cleaner (revised Fig. 3d). It is clear that the hyperphosphorylated CLOCK species could only be observed when WT PER2 was expressed.

9- Fig. 4c needs to be biochemically validated by the experiments requested above. Fig. 5 shows mutant animals have a strong phenotype but, again, biochem experiments are needed to rule out any artifact.

Response: As stated above, based on the best structural prediction method (alphafold) currently available, it is clear that the mutations do not affect the formation a-helix of the CK1-binding domain. In addition, as we showed above, deletion of the entire domain and L730 mutation resulted in similarity PER2 decay rates. It is also very important to note that our conclusion on PER phosphorylation is confirmed both in knock-in mice and in cells. We believe that our results are much more physiological relevant than the commonly published

domain deletion studies in cells based on transient transfections.

Minor comments:

1- Please revise the following sentence (2nd paragraph, Results section) and remove the second "association" word in the first sentence.

Response: Thank you! Corrected as suggested.

2- Fig. 3c: BMAL is misspelled (reads BAML1 in the figure)

Response: Thank you! Corrected as suggested.

REVIEWER COMMENTS

Reviewer #1 (Remarks to the Author):

The authors have addressed all my concerns and comments satisfactorily. The change of title and the new measurements using metabolic cage significantly strengthen their main conclusions. I have no further questions and would recommend publication of this very interesting and novel work.

Reviewer #2 (Remarks to the Author):

The revised manuscript has been greatly improved by the additional work and text editing. This outstanding work will add significantly to our growing understanding of the molecular mechanisms that underlie mammalian circadian rhythms and begins to establish that there are conserved CK1-dependent mechanisms of repression across a number of phylogenetically distinct circadian clocks in eukaryotes. I have no substantive concerns, although there are some very easily addressable minor comments that would improve clarity and maintain consistency with published literature. I enthusiastically recommend publication!

Minor

1. On line 76 in the introduction, I suggest adding a citation of the recently published paper by the Brunner group, Marzoll et al. 2022 PNAS, which shows that anchoring of CK1 to the CKBD is important for targeting non-consensus phosphorylation sites on PER2 (e.g., S662 in the FASP region and S480 in the degron). The same point is also made on line 182.
2. Use of the phrase "On the other hand, ..." (line 87) seems out of place, as the statement that follows this appears to build upon the conclusion made in the prior statement rather than oppose it.
3. The statements on line 111, 449, 468, and 483 should read "...PER and CRY-dependent (or CRY and PER-dependent) removal..." as it was shown to require both proteins in the papers cited here.
4. Of course, it is within your rights to rename the helical binding site identified here as the PCD (PER-CK1 docking site), although this was recently named as the second of two such motifs in the CBKD (i.e., CKBD-A and B) in a recent Mol Cell review by Narasimamurthy and Virshup. One wonders whether we really need a new acronym/name for this site?
5. The asterisks denoting statistical significance in the panels of Fig. 7b are quite small and hard to see.
6. While the authors present clear evidence for a highly stable interaction between the mutant PER2 and CRY1 supporting their conclusion that this complex is present at CLOCK/BMAL1-bound E-boxes, in vivo data from Koike et al. 2012, Science showed that WT PER proteins are not present to an appreciable degree from CT0-4 in CLOCK/BMAL1-bound complexes in mice. Therefore, it might be worth adding an asterisk to the PER2 cartoon in the upper right corner of Fig. 7c to indicate that this might be specific to the engineered CK1-independent mutant studied here. This caveat should also be mentioned on line 517 and in the legend when this figure panel is described.
7. The authors may want to make a speculative statement on line 475 about the long period in the Per2m/m mice; conceivably, the reduced CK1 binding they demonstrate (i.e., to PER1 only) could underlie slower kinetics of CLOCK phosphorylation and/or removal of the complex to DNA in the early repression phase to influence period. It seems rather unsatisfactory not to mention the ~4 hr lengthening in period at all here.
8. The statement on line 544 should be modified to read "...feedback loop may also be capable of..." because rhythms of Cry1/2 KO mice outside of the networked SCN are of very low amplitude, irregular period, and not sustainable. I do not think it is appropriate to make the following statement on lines 545-546 referring to "this feedback loop", insinuating that the period of the Per2m/m relies on a CRY-independent role for PER-CK1 complexes, as there are no data to support this conclusion in the manuscript and robust recruitment of PER-CK1 to CLOCK/BMAL1 has shown to depend on CRYs in Cao et al. and other work by the Sancar lab.

Reviewer #3 (Remarks to the Author):

Regarding authors' responses:

Comment 1: Ok

Comment 2: the authors refer to Fig 2F as the answer to this reviewer's question. Fig. 2F legend states: "A diagram showing the model that CK1 uses PER as scaffold to phosphorylate different PER proteins in the complex." The authors DO NOT provide the information requested by the reviewer. They just cartooned the interaction.

This reviewer asked for the structure prediction for all three proteins and the validation results of the prediction for analysis, none of which were provided.

Comment 3: The authors DO NOT provide the reviewer with the experiment requested. Instead, they run a simulation. The point the authors try to make in this paper is too important for the field to leave the ultimate decision to a simulation. In addition, the argument that a colleague who is a HHMI investigator suggested them to do that is pointless. Scientists should base their conclusions on scientific evidences coming from experiments, not other's hunches. The authors need to perform the experiment requested.

Comment 4 (Part A): The authors' answer is troublesome. Because the original experiment shown in Fig. 2A prompted questions from this reviewer, the authors present a different triplicate that show the expected result based on the reviewer's question. Shouldn't the original experiment be representative of the result? Although the new experiment proof the reviewer's request, it is worrisome how experiments are selected.

Comment 4 (Part B): The authors show a half-life experiment for PER2 and mutants. However, how was this experiment done? There is no description of the experiment in the paper, please provide this information.

Comment (e) from the reviewer original submission was NOT ANSWERED by the authors. The original comment read:

e-Why didn't the authors perform the experiment in Fig. 2C using the Delta729-738 form? This reviewer would like to see a comparison between the half-life of PER2 and the delta form when co-transfected with CK1.

Comment 6: The authors claim the antibody is not sensitive to detect endogenous CLOCK but has no problem detecting HA-CLOCK. This seems a serious technical issue that needs to be worked out.

Comment 7: OK

Comment 8: The authors state that the quality of their original experiment was not the best and now replace it by another that shows the expected result. I found this very troublesome. The authors cherry pick the experiments. They don't seem to have a quality check criterion and once a concern is raised, then they switch the blots to another set.

Comment 9: Again, the author DO NOT perform the requested biochemical validation.

Minor comments: Ok

Reviewer #1

The authors have addressed all my concerns and comments satisfactorily. The change of title and the new measurements using metabolic cage significantly strengthen their main conclusions. I have no further questions and would recommend publication of this very interesting and novel work.

Response: We appreciate the suggestions and support of this reviewer.

Reviewer #2

The revised manuscript has been greatly improved by the additional work and text editing. This outstanding work will add significantly to our growing understanding of the molecular mechanisms that underlie mammalian circadian rhythms and begins to establish that there are conserved CK1-dependent mechanisms of repression across a number of phylogenetically distinct circadian clocks in eukaryotes. I have no substantive concerns, although there are some very easily addressable minor comments that would improve clarity and maintain consistency with published literature. I enthusiastically recommend publication!

Response: We appreciate the constructive suggestions and support of this reviewer.

Minor

1. On line 76 in the introduction, I suggest adding a citation of the recently published paper by the Brunner group, Marzoll et al. 2022 PNAS, which shows that anchoring of CK1 to the CKBD is important for targeting non-consensus phosphorylation sites on PER2 (e.g., S662 in the FASP region and S480 in the degron). The same point is also made on line 182.

Response: Revised as suggested.

2. Use of the phrase “On the other hand, ...” (line 87) seems out of place, as the statement that follows this appears to build upon the conclusion made in the prior statement rather than oppose it.

Response: Change to “In addition,”.

3. The statements on line 111, 449, 468, and 483 should read “...PER and CRY-dependent (or CRY and PER-dependent) removal...” as it was shown to require both proteins in the papers cited here.

Response: Revised as suggested.

4. Of course, it is within your rights to rename the helical binding site identified here as the PCD (PER-CK1 docking site), although this was recently named as the second of two such motifs in the CBKD (i.e., CKBD-A and B) in a recent Mol Cell review by Narasimamurthy and Virshup. One wonders whether we really need a new acronym/name for this site?

Response: Because this helical domain is identified as the PER2 docking site for CK1, we prefer to keep its name as PCD.

5. The asterisks denoting statistical significance in the panels of Fig. 7b are quite small and hard to see.

Response: Revised as suggested.

6. While the authors present clear evidence for a highly stable interaction between the mutant PER2 and CRY1 supporting their conclusion that this complex is present at CLOCK/BMAL1-bound E-boxes, in vivo data from Koike et al. 2012, Science showed that WT PER proteins are not present to an appreciable degree from CT0-4 in CLOCK/BMAL1-bound complexes in mice. Therefore, it might be worth adding an asterisk to the PER2 cartoon in the upper right corner of Fig. 7c to indicate that this might be specific to the engineered CK1-independent mutant studied here. This caveat should also be mentioned on line 517 and in the legend when this figure panel is described.

Response: The reviewer is right that “WT PER proteins are not present to an appreciable degree from CT0-4 in CLOCK/BMAL1-bound complexes”, which is why we did not depict PER association with CLOCK-BMAL on DNA on the left side (Subjective Day) of the model in Figure 7c. PER-CRY was only depicted to be with CLOCK-BMAL1 on DNA in the subjective night when their levels are high. We think the model is consistent with currently published data. We have now revised the figure legend of 7c to clarify this point.

7. The authors may want to make a speculative statement on line 475 about the long period in the Per2m/m mice; conceivably, the reduced CK1 binding they demonstrate (i.e., to PER1 only) could underlie slower kinetics of CLOCK phosphorylation and/or removal of the complex to DNA in the early repression phase to influence period. It seems rather unsatisfactory not to mention the ~4 hr lengthening in period at all here.

Response: As suggested, we now added a statement in the end of this paragraph.

8. The statement on line 544 should be modified to read “...feedback loop may also be capable of...” because rhythms of Cry1/2 KO mice outside of the networked SCN are of very low amplitude, irregular period, and not sustainable. I do not think it is appropriate to make the following statement on lines 545-546 referring to “this feedback loop”, insinuating that the period of the Per2m/m relies on a CRY-independent role for PER-CK1 complexes, as there are no data to support this conclusion in the manuscript and robust recruitment of PER-CK1 to CLOCK/BMAL1 has shown to depend on CRYs in Cao et al. and other work by the Sancar lab.

Response: Agree and revised as suggested.

Reviewer #3:

Comment 2: the authors refer to Fig 2F as the answer to this reviewer’s question. Fig. 2F legend states: “A diagram showing the model that CK1 uses PER as scaffold to phosphorylate different PER proteins in the complex.” The authors DO NOT provide

the information requested by the reviewer. They just cartooned the interaction. This reviewer asked for the structure prediction for all three proteins and the validation results of the prediction for analysis, none of which were provided.

Response: In our previous response to reviewers' comments, we stated that "As suggested, we have added the alphafold structure predictions of this region with and without the mutations in Figure 1f." We think this reviewer misread Figure 1f as Fig 2f. In Figure 1f, alphafold structure predictions are presented. The L730 residue in the hPER2 was indicated by a red arrow in an enlarged picture on the left. The alphafold structure prediction is a revolutionary structure prediction method based on artificial intelligence. Its accuracy in protein structure prediction is far superior than all previous structure prediction methods. As shown in the predicted structures (below), their mutations did not affect the alpha helical structure of this region. Also importantly, amino acids 729 and 730 (colored by blue) are located at the end of a long alpha helix, thus their amino acids composition will not affect the formation of this alpha helix.

In addition, we now created hPER2 729-730AA and 730A mutants as requested by this reviewer. As shown by two independent experiments below, these mutations resulted in dramatic decreases of hPER-CK1 association when they were co-transfected in HEK293 cells (See IP:GFP and IP:FLAG below). Thus, these and 730G mutation results indicate that these residues are critical for PER-CK1 interaction.

Furthermore, as we presented in our previous response letter, we compared the PER stability of Delta729-738, PER2 GG and PER2730G. The results showed that there were no statistical differences of their decay rates (see below) and all were much more stable than the WT PER2. Together, these results demonstrate that the loss of PER-CK1 interaction in the mutants is not due to the loss of this alpha helix.

Comment 3: The authors DO NOT provide the reviewer with the experiment requested. Instead, they run a simulation. The point the authors try to make in this paper is too important for the field to leave the ultimate decision to a simulation. In addition, the argument that a colleague who is a HHMI investigator suggested them to do that is pointless. Scientists should base their conclusions on scientific evidences coming from experiments, not other's hunches. The authors need to perform the experiment requested.

Response: Please see above.

Comment 4 (Part A): The authors' answer is troublesome. Because the original experiment shown in Fig. 2A prompted questions from this reviewer, the authors present a different triplicate that show the expected result based on the reviewer's question. Shouldn't the original experiment be representative of the result? Although the new experiment proof the reviewer's request, it is worrisome how experiments are selected.

Response: The purpose of this experiment is to determine whether the 730GG mutation affects PER2 phosphorylation profile by CK1. This reviewer was asking why the amount of Per2730GG is lower in lane 5 (with CK1 expression) than in hPER2GG in lane 4 (without CK1). The original and current experimental results are both representative of the results in terms of the hPER2 phosphorylation profile, which is the only purpose of the experiment.

Since this reviewer is worried that we were cherry-picking our results, we now provide results from SIX additional experiments below, in which hPER2GG/G mutants were expressed with/without CK1. In addition, we also compared the levels of the PER730GG with/without CK1 from these experiments. It is clear that their levels are not statistically different.

Comment 4 (Part B): The authors show a half-life experiment for PER2 and mutants. However, how was this experiment done? There is no description of the experiment in the paper, please provide this information.

Response: The following information were previously provided in the Methods: “To determine hPER2 stability, PER2 and CK1 plasmids were transfected into HEK293T cells. After 36 hr, the cells were treated with 10 μg/ml CHX (Sigma). Transfected cells were collected from 0h to 8h after CHX treatment. RIPA buffer with 1x protease inhibitor cocktail and 1x phosphatase inhibitor cocktail were used to make whole cell extracts.”

Comment (e) from the reviewer original submission was NOT ANSWERED by the authors. The original comment read: e-Why didn't the authors perform the experiment in Fig. 2C using the Delta729-738 form? This reviewer would like to see a comparison between the half-life of PER2 and the delta form when co-transfected with CK1.

Response: The results requested were previously provided in second part of our response to comments 4. Since this reviewer missed it, we now copied it here again. "As suggested, we compared the PER stability of Delta729-738, PER2 GG and PER2730G. The results showed that there were no statistical differences of their decay rates (see below) and all were much more stable than the WT PER2."

Comment 6: The authors claim the antibody is not sensitive to detect endogenous CLOCK but has no problem detecting HA-CLOCK. This seems a serious technical issue that needs to be worked out.

Response: As with most transfection experiments, the level of the transgene is overexpressed at a much higher level than that of the endogenous gene. Because the endogenous CLOCK protein was expressed at a much lower level than the HA-CLOCK protein expressed from the transgene, our antibody was not able to detect it at the exposure we normally used. When a longer exposure was used, the endogenous CLOCK protein can be detected. Below, we compared the endogenous CLOCK protein levels from three different cell lines, liver and 293T cells with CLOCK transgene. As expected, the HA-CLOCK protein expression level from the transgene in 293T cells was much higher than that of the endogenous CLOCK protein. The endogenous CLOCK in the 293T cells could only be detected with a long exposure. Therefore, this is not a technical issue.

Comment 8: The authors state that the quality of their original experiment was not the best and now replace it by another that shows the expected result. I found this very troublesome. The authors cherry pick the experiments. They don't seem to have a quality check criterion and once a concern is raised, then they switch the blots to another set.

Response: Although we welcome any constructive comments and suggestions that are based on sound science and reasonable expectations, we absolutely disagree with this reviewer on his/her characterization of our study. We NEVER CHERRY PICK our experimental results and always have a high-quality control in our studies to make sure our results are reliable to the best of our efforts.

The experiment of interest concerns the hyperphosphorylation of CLOCK protein induced by PER expression. The exposure of the original data is a bit high and the proteins were a bit overloaded, which might have made the hyperphosphorylated CLOCK species less obvious to see for some. Nonetheless, to our eyes, the results were still clear to show the effect of PER expression on CLOCK hyperphosphorylation. To show that we did not cherry pick our data, we now presented results from SEVEN additional experiments below, in which we compared CLOCK phosphorylation profiles for the WT and mutant CLOCK proteins expressed from indicated transgenes. It is clear that our results are consistent in all of these experiments.

Comment 9: Again, the author DO NOT perform the requested biochemical validation.

Response: Please see above.

REVIEWERS' COMMENTS

Reviewer #3 (Remarks to the Author):

I appreciate the authors took the time to repeat the experiments and show those results to validate their conclusions. Reproducibility is of utmost importance in situations in which major statements, like the one in the paper, are stated.

The experiment with the double AA mutant has been provided and was the major concern of this reviewer. I have no problem with the publication of the article.